# FLoC: Facility Location-Based Efficient Visual Token Compression for Long Video Understanding

**Janghoon Cho**[1]**, Jungsoo Lee**[1]**, Munawar Hayat**[1]
**Kyuwoong Hwang**[1]**, Fatih Porikli**[1]**, Sungha Choi**[2†]

[1]Qualcomm AI Research*  [2]Kyung Hee University

## Abstract

Recent studies in long video understanding have harnessed the advanced visual-language reasoning capabilities of Large Multimodal Models (LMMs), driving the evolution of video-LMMs specialized for processing extended video sequences. However, the scalability of these models is severely limited by the overwhelming volume of visual tokens generated from extended video sequences. To address this challenge, we propose FLoC, an efficient visual token compression framework based on the facility location function, a principled approach that swiftly selects a compact yet highly representative and diverse subset of visual tokens within a predefined budget on the number of visual tokens. By integrating the lazy greedy algorithm, our method achieves remarkable efficiency gains by swiftly selecting a compact subset of tokens, drastically reducing the number of visual tokens while guaranteeing near-optimal performance. Notably, our approach is training-free, model-agnostic, and query-agnostic, providing a versatile solution that seamlessly integrates with diverse video-LLMs and existing workflows. Extensive evaluations on large-scale benchmarks, such as Video-MME, MLVU, LongVideoBench, and EgoSchema, show that our framework consistently surpasses recent compression techniques, highlighting its effectiveness and robustness in addressing the challenges of long video understanding as well as its processing efficiency.

## 1 Introduction

With the recent emergence of Large Language Models (LLMs) in natural language processing, there has been a surge of interest in extending their capabilities to the visual domain (Achiam et al., 2023). By utilizing the visual embeddings as token inputs to the LLMs, referred to as visual tokens, these Large Multimodal Models (LMMs) have already demonstrated their performances surpassing human-level accuracy on vision tasks, such as visual question answering (Liu et al., 2024; Fang et al., 2024; Team et al., 2023). More recently, the research focus has shifted towards enabling these models to understand video sequences (Lin et al., 2023), giving rise to video-LMMs (Song et al., 2024; Xue et al., 2024; Wang et al., 2024a; Balazevic

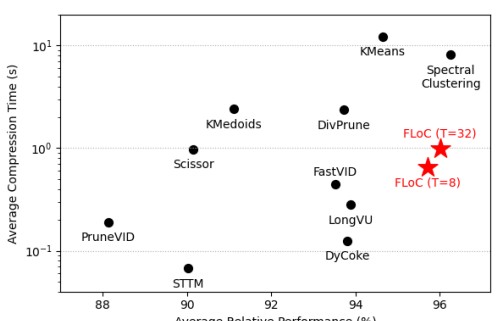

Figure 1: Performance (Average relative accuracy compared to full token usage) versus compression time (log-scale) for a number of compression algorithms. Details are described in Section 4.

et al., 2024). Such models not only excel in tasks like captioning (Krishna et al., 2017; Xu et al., 2015; Vinyals et al., 2015), event detection (Xu et al., 2019; Shou et al., 2021), and action recognition (Zhao et al., 2017; Simonyan & Zisserman, 2014), but also show significant potential in various real-world

---

*Qualcomm AI Research is an initiative of Qualcomm Technologies, Inc.
†Corresponding author. Work done while at Qualcomm AI Research.

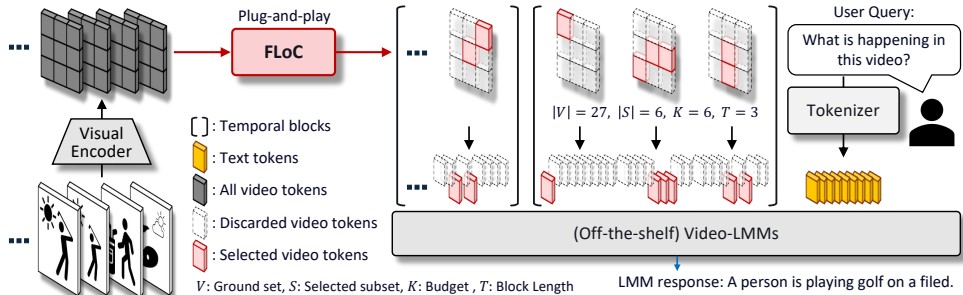

Figure 2: Overview of the proposed framework for selecting a visual token subset. Our method compresses the visual tokens extracted by a visual encoder from input video sequences into a diverse and representative subset within a given budget. The selected visual tokens are then concatenated with text tokens and fed into the video-LMM. Since our method is training-free and model-agnostic, it can be seamlessly integrated into any video-LMM in a plug-and-play manner.

applications, including surveillance through CCTV systems, immersive experiences in smart glasses, and autonomous navigation for mobile robots.

Despite this progress, long video understanding remains particularly challenging due to the explosive growth in the number of visual tokens as the video sequence length increases (Xue et al., 2024; Fu et al., 2024). When dealing with high-resolution or long-duration videos (e.g., 4K content), it becomes computationally infeasible to process every token end-to-end, especially given that most LLM-based architectures support input contexts of only 4K to 32K tokens. This limitation is exacerbated in real-world scenarios: for instance, continuous CCTV footage can span days or weeks, smart glasses may capture extended, first-person video streams, and mobile robots frequently operate in dynamic environments requiring real-time video analysis. Consequently, the gap between human-level performance and current model capabilities still exists, highlighting the complexity and significance of this research direction.

To tackle the issue of handling long video sequences, *visual token compression* is indispensable. In practice, when examining consecutive frames of a video, many tokens share highly redundant information unless there is a substantial scene change (Potapov et al., 2014). Eliminating these redundancies often does not harm the downstream performance, while excessively pruning tokens could lead to the loss of critical information. It is therefore critical to strike a delicate balance in token compression to minimize information loss.

Previous approaches to selecting representative visual tokens often relied on filtering out temporally redundant tokens or frames (Shen et al., 2024; Tao et al., 2024) or clustering techniques to extract representative information from each cluster (Wang et al., 2024b; Shang et al., 2024; Zhang et al., 2024a). While these methods may work at a reasonable level, they often fall short in capturing the full diversity needed to interpret complex visual scenes. Consider a scenario where a user wearing smart glasses searches for car keys in a cluttered room. Visual tokens representing the small object of interest (the keys) occur infrequently and sparsely within the video sequence, whereas tokens depicting general scenery, such as furniture or background, appear repeatedly and redundantly. In this setting, clustering-based approaches are likely to fail in capturing rare but important tokens—such as those corresponding to the keys—since they primarily focus on densely populated regions in the feature space. Therefore, a visual token compression algorithm that simultaneously ensures representativeness and diversity is essential to effectively retain these critical but sparse visual cues.

In order to overcome these limitations, we propose a novel visual token compression algorithm based on the *facility location* function (Lin & Bilmes, 2011; Lin et al., 2009). Our approach interprets token selection through the lens of submodular optimization, ensuring that the selected set of tokens covers all original tokens under a given budget constraint. Specifically, each subset considers the similarity between its subset and the entire tokens, enabling to include diverse information of the entire video sequence. While finding the optimal subsets in this manner is known to be a NP-hard problem, we sidestep the computational overhead by utilizing the lazy greedy algorithm (Minoux, 1978), enabling to select the visual tokens with minimal computational overheads. As a result, the chosen tokens are both representative and diverse, effectively preserving essential information for video understanding tasks. Our experiments on benchmarks such as Video-MME, MLVU,

LongVideoBench, and EgoSchema (Fu et al., 2024; Mangalam et al., 2023; Zhou et al., 2024; Wu et al., 2025) demonstrate the superiority of our method over existing approaches.

The remainder of this paper is organized as follows. In Section 2, we provide a comprehensive review of related work. Section 3 details our proposed facility location-based algorithm for visual token compression. Experimental settings and results are presented in Section 4, and we conclude in Section 5 by summarizing our key findings and discussing potential future directions.

## 2 RELATED WORK

**Sampling / Pooling** A common and straightforward strategy to deal with the abundance of visual tokens in long video sequences is to reduce the input size via pooling or sampling (Potapov et al., 2014; Cai et al., 2024; Qu et al., 2024; Wu, 2024). For instance, uniform sampling of frames or pooling across spatial/temporal dimensions can substantially cut down the computational overhead and memory usage. However, these methods often ignore the semantic importance of certain frames or regions. Such a *one-size-fits-all* approach may discard critical cues or overly compress redundant segments, leading to suboptimal performance when higher-level understanding of video content is required.

**Clustering** Another widely studied line of research involves clustering techniques to group similar frames or tokens and select representative exemplars (de Avila et al., 2011; Khosla et al., 2013; Wang et al., 2024b; Shang et al., 2024; Zhang et al., 2024a). By partitioning the visual space into clusters, these methods attempt to capture the overall distribution of the video content, retaining only the most "central" examples in each cluster. While clustering can better preserve representativeness than naive sampling, it can still struggle to guarantee coverage of rare but potentially important events. Moreover, the offline clustering process may be computationally expensive, especially for long videos, and is typically not optimized in an end-to-end manner, which can result in mismatches between clustering objectives and downstream video understanding tasks.

**Query-Aware Compression** In query-aware or task-specific compression, the aim is to select those frames or tokens that are most relevant to a given query, user interest, or downstream task (Zhang et al., 2016; Shen et al., 2024; Korbar et al., 2024; Wang et al., 2024b). This category of methods can effectively reduce the search space by focusing on what is deemed important. However, they require prior knowledge of the query or task, making them less flexible for general-purpose or zero-shot scenarios. When the query space expands or changes, such approaches often need retraining or redesign, limiting their applicability in dynamic environments (e.g., surveillance systems, smart glasses, or robots) where the set of possible queries is not fixed.

**Retraining** Learnable compression algorithms employ neural networks to decide which tokens or frames to discard or keep (Zhang et al., 2025; Argaw et al., 2024; Lee et al., 2025). By training end-to-end, they can theoretically capture complex patterns and adapt to different tasks. Nonetheless, these methods tend to require large labeled datasets and substantial training time. They are also dependent on model architecture and specific training objectives, which makes them less *model-agnostic*. Consequently, deploying such methods in rapidly evolving research fields or on resource-constrained platforms (e.g., embedded systems in mobile robots) can be challenging.

In contrast to the above approaches, our method operates in a *training-free*, plug-and-play fashion, allowing it to be easily integrated into existing pipelines with minimal overhead. Built on the principle of facility location (Lin & Bilmes, 2011; Lin et al., 2009), it interprets token selection as a submodular optimization problem, ensuring both representativeness and diversity under a given budget constraint. Additionally, we adopt a lazy greedy algorithm that significantly reduces computation time while maintaining near-optimal performance (Minoux, 1978). By decoupling the compression strategy from the underlying vision model, our approach remains *model-agnostic*, thus enabling seamless deployment in various real-world scenarios, from large-scale video analytics to on-device processing for surveillance, smart eyewear, and mobile robots. Moreover, our proposed approach operates in a *query-agnostic* manner, independent of user input. Unlike query-aware methods that require recompression for each incoming query and must retain all uncompressed tokens in memory, our method performs a one-time compression and stores only the compressed tokens. This leads to significant gains in both computational and memory efficiency.

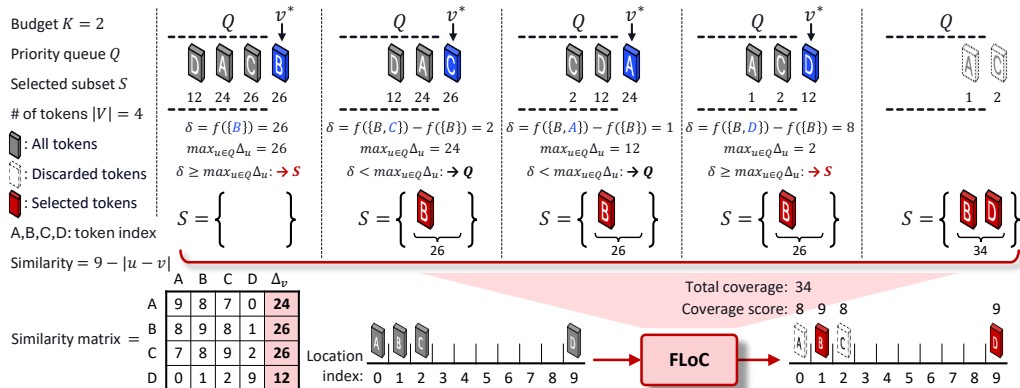

Figure 3: Illustration of the proposed algorithm for selecting a subset of visual tokens using the lazy greedy approach. The process iteratively selects tokens with the highest marginal gain while ensuring diversity and representativeness within a given budget K. This figure demonstrates the execution of Algorithm 1 from line 7 to line 14 on a one-dimensional toy example.

As demonstrated in Figure 1, our proposed method, FLoC, empirically outperforms both previously proposed approaches and traditional clustering-based methods in terms of accuracy and processing speed. This highlights its effectiveness in addressing the token compression challenge for long video understanding.

# 3    PROPOSED METHOD: FLoC

This section introduces our proposed method, *FLoC*, which employs the facility location function to select representative and diverse visual tokens. Section 3.1 outlines the overall framework, where visual tokens serve as inputs for video LMMs to generate responses. Section 3.2 then describes the facility location function and its efficient implementation using the lazy greedy alogrithm.

## 3.1    FRAMEWORK FOR VISUAL TOKEN SUBSET SELECTION

Let $V = \{v_1, v_2, \ldots, v_n\}$ be the *ground set* of all visual tokens extracted from an input video. Each token $v_i$ corresponds to a feature vector that represents a specific spatiotemporal segment (*e.g.*, a frame patch at a given time). Our goal is to select a subset $S \subseteq V$ such that $|S| \leq K$, where $K$ is a budget on the number of tokens to keep. Formally, we want to find the subset $S$ that maximizes a utility (or coverage) function $f$:

$$S^* = \arg\max_{S \subseteq V, |S| \leq K} f(S),$$

where, $f(S)$ quantifies how well the subset $S$ collectively represents or covers the entire set $V$. Specifically, $f$ should reward the chosen visual tokens (*i.e.*, $S$) that preserve the essential information and diversity of all visual tokens (*i.e.* $V$), while respecting the budget constraint $K$. Therefore, the key is to design and optimize a suitable function $f$ that captures the core video content with minimal redundancy.

The input video is first parsed into a large set of tokens, from which our method selects a representative and diverse subset. Although our method can be directly applied to the entire set of visual tokens, we divide the input video into smaller temporal blocks for computational efficiency, as shown in Figure 2. This design naturally allows future extension to streaming scenarios, where the algorithm could process accumulated tokens in a buffer. After selecting visual tokens within each block, the chosen subset is concatenated with a user-provided text prompt to form the final input for the video-LMMs. This integration seamlessly combines crucial visual cues with linguistic context, enabling the LMM to perform downstream tasks such as captioning, question answering, or event detection with improved efficiency and accuracy.

## 3.2    SUBMODULAR FACILITY LOCATION FUNCTION

We utilize the facility location function (Lin & Bilmes, 2011; Lin et al., 2009), a widely adopted submodular function, to select a representative and diverse subset of visual tokens. Formally, given a

ground set $V$ of visual tokens, the facility location objective is defined as follows:

$$f(S) = \sum_{v \in V} \max_{u \in S} \text{sim}(v, u),$$

where $\text{sim}(v, u)$ denotes the similarity between tokens $v$ and $u$. In this work, we employ cosine similarity between token embeddings as our similarity measure:

$$\text{sim}(v, u) = \frac{v^\top u}{\|v\| \|u\|}.$$

The motivation for adopting the facility location function stems from its effectiveness in balancing representativeness and diversity, making it one of the traditional and widely-used approaches for summarization tasks. By maximizing this function, the selected subset is encouraged to cover all tokens in the original set as comprehensively as possible, while avoiding redundancy by penalizing highly overlapping selections. Due to this property, facility location has been successfully applied across various summarization domains, including document summarization and video summarization tasks.

Finding an optimal subset that maximizes the facility location function is known to be NP-hard. To address this complexity, a common approximation method is the greedy algorithm, which iteratively selects tokens with the highest marginal gain until the budget constraint is satisfied. This greedy selection method guarantees a solution with a performance lower bound of $(1 - 1/e) \approx 0.632$ relative to the optimal solution (Nemhauser et al., 1978). Specifically, the greedy algorithm incrementally adds the token that provides the largest increase in coverage at each iteration.

---

**Algorithm 1** Lazy Greedy Algorithm for FLoC

---

**Require:** Ground set $V$, budget $K$
**Ensure:** Selected subset $S$ with $|S| \leq K$
1: $S \leftarrow \emptyset$
2: Initialize priority queue $Q \leftarrow \emptyset$
3: **for** $v \in V$ **do**
4: $\quad \Delta_v \leftarrow f(\{v\})$
5: $\quad$ Insert $v$ into $Q$ with priority $\Delta_v$
6: **end for**
7: **while** $|S| < K$ **do**
8: $\quad v^* \leftarrow \arg\max_{v \in Q} \Delta_v$ (pop from queue)
9: $\quad \delta \leftarrow f(S \cup \{v^*\}) - f(S)$
10: $\quad$ **if** $\delta \geq \max_{u \in Q} \Delta_u$ **then**
11: $\quad\quad S \leftarrow S \cup \{v^*\}$
12: $\quad$ **else**
13: $\quad\quad$ Update priority of $v^*$ in $Q$ to $\delta$ and re-insert
14: $\quad$ **end if**
15: **end while**
$\quad$ **return** $S$

---

To further enhance computational efficiency, we implement a lazy greedy algorithm (Minoux, 1978), which significantly reduces the computational overhead by avoiding unnecessary recomputation of marginal gains. Specifically, the algorithm exploits the submodularity (diminishing returns) property of the facility location function $f$. Formally, for any subsets $A \subseteq B \subseteq V$ and a token $v \in V \setminus B$, the marginal gain satisfies:

$$f(A \cup \{v\}) - f(A) \geq f(B \cup \{v\}) - f(B)$$

This inequality implies that the marginal benefit of adding a visual token $v$ can only decrease or remain constant as the selected subset grows. Consequently, the marginal gain computed in a previous iteration serves as a valid upper bound for the current marginal gain. We leverage this by maintaining a priority queue of these upper bounds. In each step of the search process, we pop the candidate $v^*$ with the highest upper bound and recompute its exact marginal gain $\delta$ with respect to the current subset. If $\delta$ remains greater than or equal to the upper bounds of all other candidates in the queue, submodularity guarantees that $v^*$ is the optimal choice for the current iteration without needing to re-evaluate the rest. Algorithm 1 outlines the detailed procedure, and Figure 3 provides a visual illustration of this process.

The lazy greedy algorithm significantly reduces computational complexity compared to the naive greedy approach. While the naive greedy algorithm for maximizing submodular functions has a time complexity of $O(nK)$, the lazy greedy approach leverages the submodularity property to avoid unnecessary recomputation of marginal gains. By using a priority queue, it updates marginal gains only when needed, achieving empirical speedups often approaching an order of magnitude. Consequently, it becomes particularly efficient for handling numerous visual tokens and enabling real-time processing of long videos.

Compared to traditional clustering-based methods, our lazy greedy-based facility location method offers several advantages. First, it eliminates iterative refinement and costly operations such as

eigen-decompositions. Instead, our approach directly selects tokens in a single forward pass by maximizing global coverage, ensuring a diverse and representative subset is chosen efficiently. Thus, it provides a highly efficient and scalable alternative, especially suitable for real-time or on-device processing requirements. Next, the facility location function explicitly optimizes global coverage by selecting tokens that best represent the entire set of visual tokens. Unlike $k$-means, which tends to select tokens from dense regions and may overlook sparsely populated yet important regions (*e.g.*, *rare objects like keys, subtle actions, or fine-grained details such as small text or facial expressions*), our method ensures that selected tokens span diverse feature regions by defining utility in terms of coverage, prioritizing selections that maximize representativeness. This prevents oversampling from dense clusters while preserving rare but meaningful patterns.

In our empirical evaluation, we observed that the proposed lazy greedy-based facility location algorithm significantly outperforms traditional clustering methods, such as $k$-means and spectral clustering, in terms of computational efficiency. Specifically, our experiments demonstrate substantial runtime improvements, achieving speedups of several times or more depending on the dataset size and scenario. We provide detailed experimental results and analysis comparing the runtime performance of our method against other clustering baselines in Section 4.

## 4 EXPERIMENTS

### 4.1 MODELS

**Qwen2.5-VL** (Bai et al., 2025) is an advanced vision-language model capable of handling high-resolution images and long video sequences. It introduces dynamic resolution processing via a Window Attention-based Vision Transformer and supports absolute temporal encoding.

**InternVL3** (Zhu et al., 2025) is a multimodal model designed with native vision-language pretraining and Cascade Reinforcement Learning. For long video understanding, it incorporates a Visual Resolution Router to dynamically allocate visual token capacity across frames.

**Others.** We also conducted experiments on **Qwen2-VL** (Wang et al., 2024a) and **LLaVA-Next-Video** (Zhang et al., 2024c) models to further validate the generalizability of our approach. Due to space limitations, detailed results and analysis for these models are provided in the Appendix. [1]

### 4.2 BENCHMARKS

**Video-MME** (Fu et al., 2024) is a multi-modal evaluation benchmark designed to assess visual and textual understanding in videos, covering diverse real-life footage across domains such as sports, news, and user-generated content. It focuses on tasks like video captioning, event detection, and question answering.

**LongVideoBench** (Wu et al., 2025) is curated for long-form video understanding, featuring extended videos such as lectures, live events, and surveillance footage, emphasizing topic segmentation and global summarization.

**MLVU** (Multi-Level Video Understanding) (Zhou et al., 2024) evaluates hierarchical comprehension from frame-level recognition to storyline interpretation, using clips from movies, documentaries, and instructional videos. [2]

**EgoSchema** (Mangalam et al., 2023) evaluates egocentric video understanding through short, first-person perspective clips, emphasizing schema-level reasoning and activity prediction.

Among these, **Video-MME**, **LongVideoBench**, and **MLVU** include videos longer than one hour, making them suitable for long-form video understanding. In contrast, **EgoSchema** consists of relatively short, minute-level clips but remain widely adopted benchmarks for video understanding research.

#### 4.2.1 IMPLEMENTATION

We effectively evaluated the performance of various visual token compression algorithms using the `lmms-eval` toolkit (Li et al., 2024; Zhang et al., 2024b) as our codebase, which supports multiple

---

[1] Qwen2.5-VL, Qwen2-VL, and LLaVA-Video-7B-Qwen2 are all under the Apache-2.0 license. InternVL3 is under the MIT license.

[2] Video-MME, LongVideoBench, and MLVU are all under the CC BY-SA 4.0 International License.

Table 1: Comparison of visual token compression methods. The ratio indicates the compression ratio relative to the original number of visual tokens.

| Model | | Qwen2.5-VL-7B | | | | | InternVL3-8B | | | | |
|---|---|---|---|---|---|---|---|---|---|---|---|
| Comp. Ratio | Method | Video MME | MLVU | LVB | Ego Schema | Avg. | Video MME | MLVU | LVB | Ego Schema | Avg. |
| 1 | - | 66.33 | 70.31 | 60.51 | 61.40 | 64.64 | 66.63 | 72.68 | 59.39 | 70.00 | 67.18 |
| $2^{-3}$ | TS-LLaVA | 61.15 | 67.57 | 55.20 | 59.60 | 60.88 | 62.78 | 67.30 | 56.02 | 68.20 | 63.58 |
| | LongVU | 62.19 | 66.61 | 55.42 | 59.40 | 60.91 | 64.70 | 69.50 | 55.35 | 69.20 | 64.69 |
| | DivPrune | 61.63 | 67.57 | 56.17 | 58.40 | 60.94 | 64.07 | 70.06 | **56.92** | 65.00 | 64.01 |
| | Random | 60.30 | 66.24 | 55.72 | 58.60 | 60.22 | 60.59 | 65.69 | 56.02 | 65.20 | 61.88 |
| | DyCoke | 62.11 | 67.53 | 55.12 | 59.60 | 61.09 | 63.96 | 68.45 | 55.72 | 69.00 | 64.28 |
| | PruneVid | 58.19 | 64.54 | 54.15 | 54.20 | 57.77 | 57.41 | 62.05 | 53.48 | 62.80 | 58.94 |
| | STTM | 59.52 | 63.57 | 54.60 | 55.80 | 58.37 | 63.52 | 64.26 | 54.90 | 66.20 | 62.22 |
| | Scissor | 58.59 | 65.04 | 54.08 | 56.40 | 58.53 | 61.15 | 67.76 | 55.12 | 65.80 | 62.46 |
| | FastVID | 60.89 | 67.31 | 57.14 | 58.60 | 60.99 | - | - | - | - | - |
| | **FLoC (Ours)** | **63.33** | **68.81** | **58.12** | **60.00** | **62.57** | **64.93** | **71.57** | 56.69 | **69.40** | **65.65** |
| $2^{-4}$ | TS-LLaVA | 58.78 | 64.67 | 52.51 | 57.20 | 58.29 | 59.63 | 64.95 | 53.85 | 62.80 | 60.31 |
| | LongVU | 58.07 | 62.97 | 52.73 | 55.40 | 57.29 | 56.48 | 60.12 | 51.31 | 60.40 | 57.08 |
| | DivPrune | 58.85 | 64.67 | 54.00 | 55.80 | 58.33 | 61.93 | 68.08 | 54.82 | 61.80 | 61.66 |
| | Random | 57.44 | 63.80 | 53.63 | **58.20** | 58.27 | 59.74 | 64.77 | 54.23 | 66.60 | 61.34 |
| | DyCoke | 57.00 | 63.02 | 53.78 | 54.00 | 56.95 | 61.37 | 65.13 | 53.10 | **67.40** | 61.75 |
| | PruneVid | 54.11 | 61.59 | 51.83 | 52.00 | 54.88 | 53.81 | 59.48 | 52.28 | 58.40 | 55.99 |
| | STTM | 57.15 | 61.73 | 51.68 | 50.80 | 55.34 | 60.15 | 62.93 | 52.28 | 63.40 | 59.69 |
| | Scissor | 55.26 | 60.95 | 53.55 | 54.00 | 55.94 | 58.89 | 64.44 | 53.77 | 63.40 | 60.13 |
| | FastVID | 58.67 | 65.52 | 54.23 | 57.20 | 58.91 | - | - | - | - | - |
| | **FLoC (Ours)** | **60.89** | **66.19** | **55.27** | 58.00 | **60.09** | **63.41** | **69.09** | **56.47** | 66.20 | **63.79** |
| $2^{-5}$ | TS-LLaVA | 55.07 | 62.37 | 50.49 | 54.60 | 55.63 | 58.89 | 63.89 | 53.33 | 61.00 | 59.28 |
| | LongVU | 53.41 | 58.42 | 50.34 | 53.20 | 53.84 | 55.96 | 59.52 | 51.01 | 58.80 | 56.32 |
| | DivPrune | 55.78 | 61.91 | 52.28 | 53.60 | 55.89 | **60.85** | 65.46 | 52.88 | 59.40 | 59.65 |
| | Random | 55.56 | 61.41 | 49.89 | 53.60 | 55.12 | 57.30 | 63.57 | 52.13 | 60.40 | 58.35 |
| | DyCoke | 54.37 | 59.98 | 51.38 | 54.60 | 55.08 | 59.22 | 62.60 | 51.98 | 63.00 | 59.20 |
| | PruneVid | 51.11 | 58.51 | 49.66 | 49.00 | 52.07 | 51.41 | 56.39 | 49.96 | 53.00 | 52.69 |
| | STTM | 55.26 | 59.25 | 50.11 | 49.40 | 53.51 | 57.52 | 61.73 | 52.43 | 60.80 | 58.12 |
| | Scissor | 51.89 | 58.74 | 51.46 | 50.00 | 53.02 | 56.33 | 61.68 | 51.31 | 62.60 | 57.98 |
| | FastVID | 57.19 | 62.94 | 52.95 | **55.00** | 57.02 | - | - | - | - | - |
| | **FLoC (Ours)** | **58.63** | **64.08** | **53.10** | 54.00 | **57.45** | 60.81 | **66.93** | **54.23** | **63.80** | **61.44** |

Table 2: Evaluation of token compression with extended temporal input (1 FPS, up to 7200 Frames)

| Model | | Qwen2.5-VL-7B | | | | Qwen2.5-VL-32B | | | |
|---|---|---|---|---|---|---|---|---|---|
| Max Frames | Method | Video MME | MLVU | LVB | Avg. | Video MME | MLVU | LVB | Avg. |
| 768 | - | **66.33** | 70.31 | 60.51 | 65.82 | 70.41 | 71.57 | 62.60 | 68.19 |
| 7200 | TS-LLaVA | 65.07 | 72.40 | 62.08 | 66.52 | 70.22 | 73.09 | 65.00 | 69.44 |
| | LongVU | 65.04 | 71.02 | 62.75 | 66.27 | 70.37 | 72.22 | 64.62 | 69.07 |
| | DivPrune | 64.93 | 70.19 | 62.30 | 65.81 | 70.26 | 73.37 | 64.32 | 69.32 |
| | Random | 64.56 | 70.52 | 61.63 | 65.57 | 69.70 | 72.49 | 64.62 | 68.94 |
| | DyCoke | 65.78 | 71.30 | **62.98** | 66.69 | 71.00 | 72.26 | 63.87 | 69.04 |
| | PruneVid | 62.96 | 68.63 | 62.45 | 64.68 | 68.00 | 70.19 | 63.50 | 67.23 |
| | **FLoC (Ours)** | 65.85 | **72.63** | 62.60 | **67.03** | 71.56 | **73.83** | **66.49** | **70.63** |

video LMM models and diverse benchmarks. All experiments were conducted leveraging NVIDIA H100 GPUs and multiprocessing for efficient computation.

## 4.3 BASELINES

**Recent Algorithms** We compared the performance of recently proposed algorithms, LongVU (Shen et al., 2024), DyCoke (Tao et al., 2024), TS-LLaVA (Qu et al., 2024), PruneVID (Huang et al., 2024), DivPrune (Alvar et al., 2025), STTM (Hyun et al., 2025), LLaVA Scissor (Sun et al., 2025), and FastVID (Shen et al., 2025). Implementation details are described in Section E of Appendix.

**Clustering Algorithms** We used K-means, K-medoids, and Spectral clustering algorithms as our baselines.

## 4.4 RESULTS

To simulate realistic deployment scenarios where memory resources are constrained—such as on-device execution of LMMs—we compress visual tokens to reduced lengths (1/8, 1/16, 1/32 of the optimal visual token number) and evaluate the models' robustness through long video understanding. This setup allows us to assess how well LMMs retain performance under severe token budget limitations. We additionally measured the compression time of each algorithm to analyze the trade-off between performance and efficiency, providing insights into their practical applicability.

Table 1 presents a comparative analysis of video understanding performance using above-mentioned 6 benchmarks with 9 different baseline methods as described in 4.3. We evaluate these methods under various visual token compression ratios of $2^{-5}$, $2^{-4}$, and $2^{-3}$. Figure 1 illustrates the performance of each compression algorithm in terms of accuracy retention (x-axis), measured as a percentage relative to the full-token baseline, and compression time (y-axis). The results are based on a 1/8 compression ratio using the Qwen2.5-VL-7B model.

As shown in the results, our method consistently outperforms existing visual token compression techniques across different datasets, compression ratios, and backbone models. We attribute this superiority to FLoC's ability to overcome the structural limitations of prior approaches. Specifically, graph-based merging methods (e.g., STTM, LLaVA Scissor) often suffer from the "weak connection" problem, where distinct tokens—such as small objects and their background—are irreversibly merged based on local similarity thresholds, leading to significant detail loss especially at low compression ratios. Similarly, while diversity-based methods (e.g., DivPrune) effectively capture outliers, they often fail to retain representative tokens that describe the core context of the video. In contrast, our facility location-based approach mathematically guarantees a balance between representativeness and diversity, successfully retaining both the central narrative and fine-grained visual cues that other methods overlook.

As shown in Figure 1, clustering-based methods such as K-Means and Spectral Clustering occasionally achieve performance comparable to our proposed approach. However, these methods incur approximately 10× higher compression time, indicating a significant disadvantage in terms of efficiency. A detailed comparison of the efficiency of clustering-based methods is provided in the following subsection.

In the final experiment, we aimed to fully leverage the optimal token length of the LMM by extracting all visual tokens from as many frames as possible in a long video sequence, and compressing them to the model's optimal token length. Specifically, we modified the default Qwen2.5-VL vision processing script—which originally supports up to 768 frames—to handle up to 7,200 frames. The resulting visual tokens were then compressed to 24,576 tokens, corresponding to the optimal token length of the model. The performance under this setting is presented in Table 2.

As shown in Table 2, FLoC can significantly improve the performance of LMMs that are conventionally measured using a limited number of frames. For the 7B model, the accuracy increased by an average of **1.21 points**, and for the 32B model, it rose by an average of **2.44 points**. These results indicate that while existing LMMs are forced to process fewer frames due to their limited context length, our proposed algorithm enables them to handle a larger number of frames through efficient compression. We believe this approach substantially enhances their overall video understanding capabilities.

These findings demonstrate that our proposed algorithm enables LMMs to generate high-quality responses under resource-constrained conditions, with significantly reduced processing time.

## 4.5 ANALYSIS

### 4.5.1 REPRESENTATIVE AND DIVERSE VISUAL TOKENS

We demonstrate the effectiveness of our method in selecting representative and diverse visual tokens through t-SNE visualization. For the visualization, we use Qwen2-VL 7B as the model and a randomly selected video in VideoMME as the dataset. We compare the projected embedding spaces obtained using K-means, K-Medoids, spectral clustering, and ours. In Fig. 4, red-colored stars and black-colored dots represent the selected and discarded visual tokens for each algorithm, respectively.

As shown, K-means and K-Medoids clustering predominantly select representative visual tokens from dense regions while failing to capture diverse tokens. In contrast, facility location selects

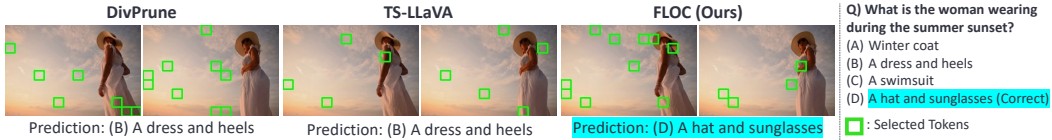

| K-means Clustering | K-medoids Clustering | Spectral Clustering | Ours |

Figure 4: TSNE visualization of visual tokens. The red-colored stars and black-colored dots indicate the selected and discarded visual tokens, respectively. As shown, our method selects both representative and diverse visual tokens.

Figure 5: FLoC captures diverse visual tokens (e.g., hat, sunglasses) missed by DivPrune and TS-LLaVA, enabling accurate answers about what the woman is wearing.

visual tokens those are evenly distributed from both major and minor clusters, ensuring a more diverse representation. This visualization clearly highlights that our proposed method effectively preserves both representative and diverse visual tokens, which are crucial for comprehensive video understanding.

Additionally, as shown in Fig. 5, our proposed FLoC selects diverse tokens, successfully capturing visual cues like hats and sunglasses, unlike DivPrune and TS-LLaVA, which often miss them. This enables more accurate answers to questions about what the woman is wearing. Additional results with more examples are provided in the Appendix, specifically illustrated in Figure 8 and 9.

We further validate that visual tokens compressed by FLoC are more representative and diverse compared to those produced by alternative compression algorithms, supported by both quantitative metrics and empirical evidence. These comparisons are visualized in Figure 7 of the Appendix, where representativeness and diversity are explicitly quantified. Moreover, as shown in Table 6 of the Appendix, our framework achieves outstanding performance on the MLVU dataset, particularly in tasks requiring fine-grained video understanding such as Needle QA and Ego Reasoning, further substantiating the superiority of our approach.

### 4.5.2 MINIMAL COMPUTATIONAL OVERHEADS

We also compare the computational overhead of our proposed method with other visual token compression techniques. We use Qwen2-VL 7B as the model and VideoMME as the dataset for the experiment. We measure the time taken by each method to perform visual token compression.

As shown in Table 3, our method consistently achieves the lowest computational cost across different numbers of the block length, denoted as $T$. Notably, the performance gap in computational efficiency between our method and clustering-based approaches widens as $T$ increases, further highlighting the scalability of our approach. Clustering-based methods, such as K-Means, K-Medoids, and spectral clustering, often incur substantial computational overhead when applied to visual token compression. For instance, K-Means requires multiple iterations to update cluster centroids until convergence, involving computations proportional to $O(nKdi)$, where $d$ denotes the dimensionality of features, and $i$ indicates the number of iterations. Although K-Medoids selects actual data points as cluster centers and may converge faster in practice, it still typically scales as $O(K(n - K)^2)$, becoming computationally intensive as $n$ grows. Similarly, spectral clustering involves expensive eigen-decomposition of similarity matrices, incurring a computational complexity of approximately $O(n^3)$ in general. These inherent limitations significantly reduce the practicality of clustering-based methods for compressing visual tokens, especially in long video sequences with extremely large token sets.

In contrast, our method circumvents these computational bottlenecks by leveraging the lazy greedy algorithm, which exploits submodularity to efficiently select a near-optimal subset of tokens. Instead of exhaustively evaluating all possible token selections, the lazy greedy approach prioritizes promising

Table 3: Comparisons of average computation times (sec).

| Methods | Time Complexity | $2^{-5}$ | | | $2^{-4}$ | | | $2^{-3}$ | | | Average Accuracy |
|---|---|---|---|---|---|---|---|---|---|---|---|
| | | $T=2$ | $T=8$ | $T=32$ | $T=2$ | $T=8$ | $T=32$ | $T=2$ | $T=8$ | $T=32$ | |
| K-Means | $O\left(n \cdot K \cdot d \cdot i\right)$ | 0.551 | 4.630 | 59.00 | 0.790 | 8.860 | 113.0 | 1.390 | 16.80 | 218.0 | 58.66 |
| K-Medoids | $O\left(K \cdot (n-K)^2\right)$ | 0.022 | 0.113 | 0.716 | 0.018 | 0.119 | 0.747 | 0.021 | 0.135 | 0.877 | 56.22 |
| Spectral Clustering | $O\left(n^3\right)$ | 0.232 | 0.569 | 5.160 | 0.794 | 2.260 | 9.650 | 0.270 | 1.180 | 21.10 | 58.97 |
| FLoC (Ours) | $O\left(n \cdot K\right)$ | **0.010** | **0.056** | **0.413** | **0.012** | **0.065** | **0.475** | **0.014** | **0.075** | **0.527** | **59.74** |

candidates while skipping redundant computations, significantly reducing the runtime. These results demonstrate that our method not only provides superior video understanding performance but also achieves minimal computational overhead, making it highly practical for real-world applications.

### 4.5.3 ROBUSTNESS ON BLOCK LENGTHS

To examine the impact of the sole hyperparameter in our proposed algorithm—the block length $T$—we evaluated performance across various datasets and compression ratios while varying $T$. In this experiment, the Qwen2VL-7B model was used.

As illustrated in Figure 6, we observe distinct behaviors depending on the block length $T$. In the region where $T \leq 4$, performance tends to degrade because the narrow temporal window prevents the algorithm from identifying redundancy across adjacent blocks (inter-block redundancy). Conversely, as $T$ increases, the facility location objective optimizes representativeness and diversity over a broader temporal context, leading to performance saturation. Crucially, unlike traditional clustering methods where computational cost scales quadratically with input size, our lazy greedy implementation ensures that increasing $T$ incurs negligible latency overhead. This suggests that a sufficiently large fixed block length (e.g., $T = 32$) serves as a robust and efficient default, minimizing the need for per-video hyperparameter tuning. However, we acknowledge that relying on fixed uniform segmentation is a heuristic simplification and may not be strictly optimal for every video content. We anticipate that developing an adaptive mechanism to automatically determine $T$ based on temporal dynamics could yield further performance improvements. A more detailed discussion on this limitation and potential future directions is provided in Appendix I

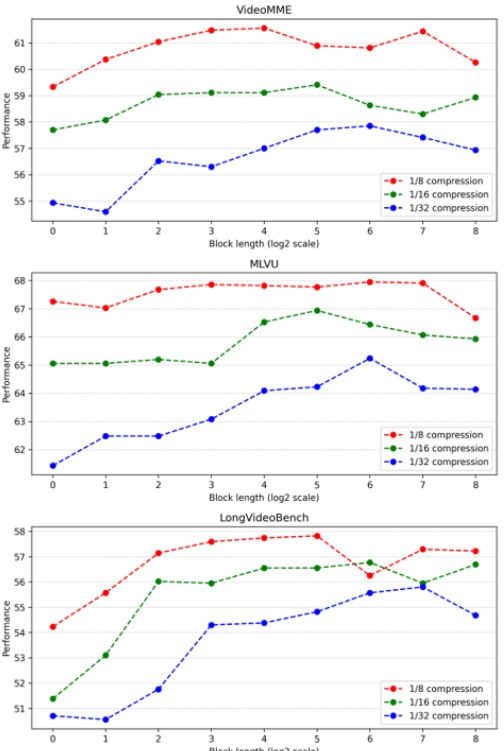

Figure 6: Performance versus block length $T$ (log2-scale) for a number of benchmark datasets.

## 5 CONCLUSION

As long video understanding advances, handling the overwhelming number of visual tokens remains a key bottleneck. While prior methods such as uniform sampling and clustering have addressed this issue, they often fail to capture sufficient visual diversity and add computational overhead. We tackle these limitations by proposing a visual token compression framework based on the facility location function. Our method selects tokens that are both representative and diverse, preserving essential scene information while significantly reducing computation via a lazy greedy algorithm. Extensive experiments on large-scale benchmarks, including Video-MME, LongVideoBench, MLVU, and Egoschema show that our method consistently outperforms existing compression techniques. Its efficiency and strong performance without added overhead make it well-suited for real-world applications such as surveillance, augmented reality, and autonomous navigation. As video-LMMs scale, improving efficiency and information retention will be key to advancing long video understanding.

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

APPENDIX

## A    REPRESENTATIVENESS AND DIVERSITY

To quantitatively verify the representativeness and diversity of our proposed facility location-based visual token selection algorithm, we conducted an analysis using two complementary metrics: (1) **averaged sum coverage**, measuring how comprehensively the selected tokens cover the entire set of visual tokens, defined as

$$\text{Averaged Sum Coverage}(S) = \frac{1}{|V||S|} \sum_{v \in V} \sum_{u \in S} \text{sim}(v, u),$$

where $V$ is the entire set of visual tokens, $S$ is the selected subset, and $\text{sim}(v, u)$ is the cosine similarity between tokens $v$ and $u$, and (2) **averaged distance**, computed as the average pairwise distance (using $1 - \text{sim}(u, w)$) among the selected tokens:

$$\text{Averaged Distance}(S) = \frac{1}{|S|(|S| - 1)} \sum_{u \in S} \sum_{w \in S, w \neq u} (1 - \text{sim}(u, w)).$$

We compared our method against three clustering-based baselines: K-means, K-medoids, and spectral clustering.

We utilized 50 randomly selected videos from the Video MME dataset and employed the Qwen2-vl 7B model. Due to the significant variability in the range of measures across different data points, we normalized the six measures obtained from six algorithms for each video to have a zero mean and a standard deviation of one. The normalized results were then visualized using a scatter plot.

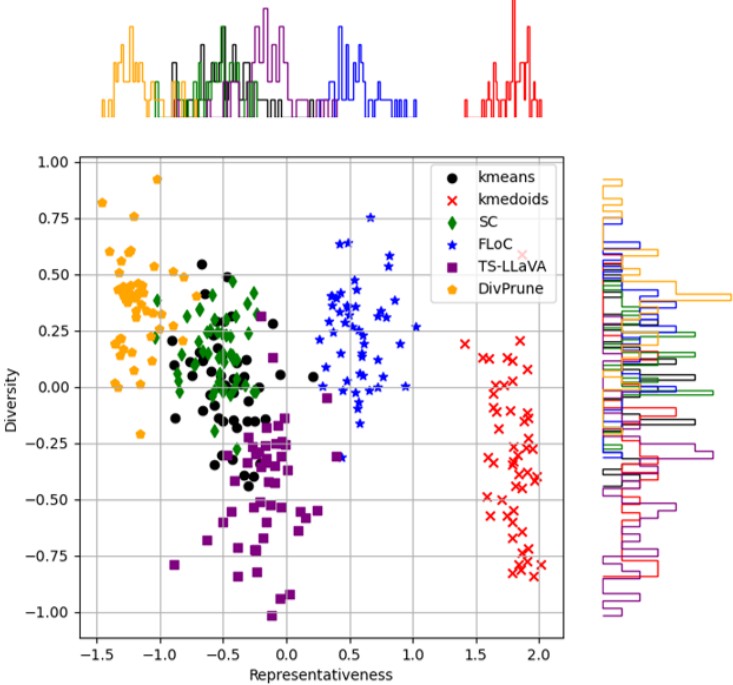

Figure 7: Scatter plot of each algorithm's representativeness and diversity.

As shown in Figure 7, our facility location approach consistently outperformed the baselines in both representativeness and diversity measures. Specifically, our method achieved higher averaged sum coverage scores, indicating superior representativeness, and greater averaged distance, demonstrating its effectiveness in selecting both representative and diverse tokens.

In the scatter plot, the values obtained using the proposed FLoC algorithm are predominantly located in the first quadrant. This indicates that, after normalization, the values are on average more representative and diverse compared to other algorithms. When compared to k-medoids, the FLoC

algorithm shows lower representativeness but superior diversity. When compared to DivPrune, our proposed algorithm shows slightly lower diversity but superior representativeness. Additionally, when compared to TS-LLaVA, k-means and spectral clustering, the FLoC algorithm demonstrates superiority in both representativeness and diversity.

These results suggest that when selecting representative samples from an entire ground set, two crucial factors to consider are representativeness and diversity, which inherently exist in a trade-off relationship. If samples are densely distributed in a specific region, selecting a disproportionately large number of samples from that area can reduce overall diversity. Conversely, focusing excessively on diversity might lead to neglecting important samples from these densely populated, and potentially critical, regions. Our proposed FLoC effectively addresses this trade-off by selecting tokens that are both representative and diverse. Consequently, FLoC achieves superior performance in long video understanding tasks.

## B COMPREHENSIVE PERFORMANCE EVALUATION

We present the performance of all evaluated visual token compression algorithms across the three benchmark datasets and three backbone LLM models in Table 4 and Table 5. In the previously submitted manuscript, results for several clustering-based methods, namely k-means, k-medoids, and spectral clustering, were omitted from the main performance tables due to space constraints. These are now included for a comprehensive comparison.

Table 4: Full comparison of visual token compression methods. Backbone LLM is LLaVA-Video-7B-Qwen2.

| Ratio | Tokens | Frames | Methods | Video-MME | | | | Long Video Bench | | | | | MLVU | Avg. |
|---|---|---|---|---|---|---|---|---|---|---|---|---|---|---|
| | | | | Short | Medium | Long | Overall | 15 | 60 | 600 | 3600 | Overall | | |
| 100% | 21632 | 128 | - | 75.78 | 63.33 | 54.67 | 64.59 | 66.67 | 68.61 | 58.98 | 51.77 | 58.27 | 70.39 | 64.42 |
| | | 16 | Frame Uniform | 68.78 | 54.78 | 49.33 | 57.63 | 54.50 | 66.38 | 54.37 | 50.00 | 54.08 | 53.66 | 50.31 |
| $2^{-3}$ | 2704 | 128 | Pooling | 65.33 | 53.89 | 48.67 | 55.96 | 56.61 | 68.61 | 56.31 | 48.05 | 54.45 | 61.24 | 57.22 |
| | | | LongVU | 68.89 | 58.44 | 51.67 | 59.67 | 56.61 | 65.12 | 55.83 | 52.31 | 55.65 | 62.57 | 59.30 |
| | | | TS-LLaVA | 71.00 | 59.56 | 50.56 | 60.37 | 57.67 | 68.02 | 58.98 | 51.60 | 56.84 | 65.15 | 60.79 |
| | | | DivPrune | 69.11 | 59.22 | 52.56 | 60.30 | 58.20 | 65.12 | 56.55 | 51.42 | 55.72 | 65.00 | 60.34 |
| | | | K-means | 71.78 | 59.22 | 50.11 | 60.37 | 60.38 | 69.93 | 60.41 | 50.89 | 57.19 | 66.59 | 61.38 |
| | | | K-medoids | 68.56 | 56.89 | 49.78 | 58.41 | 56.61 | 68.02 | 57.77 | 50.71 | 55.95 | 62.11 | 58.82 |
| | | | SC | 72.11 | 61.78 | 51.56 | **61.81** | 60.32 | 68.02 | 59.22 | 52.13 | 57.52 | 66.07 | 61.80 |
| | | | **FLoC (Ours)** | 71.68 | 60.56 | 50.89 | 61.04 | 61.91 | 69.19 | 60.19 | 51.60 | **57.97** | 67.43 | **62.15** |
| | | 8 | Frame Uniform | 60.78 | 51.89 | 48.56 | 53.74 | 43.92 | 56.40 | 53.16 | 49.82 | 50.86 | 57.20 | 53.93 |
| $2^{-4}$ | 1352 | 128 | Pooling | 60.00 | 50.11 | 45.33 | 51.81 | 54.50 | 59.30 | 50.73 | 44.86 | 49.89 | 57.56 | 53.09 |
| | | | LongVU | 62.56 | 53.78 | 47.11 | 54.48 | 51.85 | 62.21 | 51.21 | 49.65 | 52.06 | 56.74 | 54.43 |
| | | | TS-LLaVA | 67.22 | 56.78 | 50.56 | 58.19 | 56.09 | 68.02 | 58.25 | 47.52 | 54.68 | 61.52 | 58.13 |
| | | | DivPrune | 67.67 | 57.67 | 50.00 | 58.44 | 56.61 | 64.54 | 54.84 | 48.23 | 53.55 | 62.24 | 58.08 |
| | | | K-means | 69.22 | 55.67 | 50.33 | 58.41 | 59.26 | 66.28 | 58.74 | 49.11 | 55.72 | 63.08 | 59.07 |
| | | | K-medoids | 65.56 | 53.67 | 47.89 | 55.70 | 53.44 | 66.28 | 54.85 | 47.34 | 52.95 | 59.17 | 55.94 |
| | | | SC | 68.44 | 56.56 | 51.56 | 58.85 | 56.61 | 68.02 | 56.07 | 50.00 | 55.12 | 64.05 | 59.34 |
| | | | **FLoC (Ours)** | 69.33 | 57.44 | 51.00 | **59.26** | 60.32 | 68.02 | 58.74 | 48.76 | 55.95 | **64.54** | **59.92** |
| | | 4 | Frame Uniform | 52.56 | 49.44 | 44.44 | 48.81 | 43.92 | 51.16 | 51.46 | 46.99 | 48.47 | 53.66 | 50.31 |
| $2^{-5}$ | 676 | 128 | Pooling | 57.00 | 49.11 | 45.00 | 50.37 | 50.79 | 56.40 | 49.27 | 43.97 | 48.17 | 54.94 | 51.16 |
| | | | LongVU | 57.33 | 49.78 | 45.78 | 50.96 | 49.21 | 56.40 | 48.30 | 48.23 | 49.44 | 53.61 | 51.34 |
| | | | TS-LLaVA | 64.22 | 55.00 | 47.78 | 55.67 | 52.38 | 65.12 | 53.64 | 46.45 | 51.91 | 57.52 | 55.03 |
| | | | DivPrune | 64.00 | 56.22 | 49.00 | **56.41** | 53.97 | 55.81 | 51.94 | 46.10 | 50.26 | 59.43 | 55.37 |
| | | | K-means | 65.22 | 49.56 | 47.22 | 54.00 | 56.67 | 67.02 | 57.50 | 46.87 | 54.12 | 58.49 | 55.54 |
| | | | K-medoids | 63.67 | 50.67 | 47.44 | 53.93 | 51.32 | 63.37 | 55.10 | 46.28 | 51.91 | 55.82 | 53.89 |
| | | | SC | 65.00 | 54.56 | 47.89 | 55.81 | 49.74 | 63.95 | 53.88 | 48.05 | 52.13 | 59.40 | 55.78 |
| | | | **FLoC (Ours)** | 66.44 | 54.00 | 48.22 | 56.22 | 55.03 | 67.44 | 55.34 | 48.76 | **54.07** | **61.22** | **57.17** |

As evidenced by these tables, our proposed model achieves the highest average performance across all three benchmark datasets for all considered backbone LLM models and at all compression ratios. This consistent superiority indicates that our algorithm effectively selects representative visual tokens crucial for long video understanding, irrespective of the specific backbone model architecture or the nature of the question query.

## C DETAILED TASK-SPECIFIC PERFORMANCE ANALYSIS ON MLVU

To thoroughly investigate the factors contributing to the performance improvements of our proposed algorithm, we conducted a comparative analysis of its performance on seven distinct sub-tasks within the MLVU dataset. The MLVU dataset is broadly categorized into three main types of tasks: Holistic Long Video Understanding (LVU), Single Detail LVU, and Multi Detail LVU. These are further

Table 5: Full comparison of visual token compression methods. Backbone LLMs are Qwen2-VL-2B and Qwen2-VL-7B.

| Model | Ratio | Tokens | Frames | Methods | Video-MME | | | | Long Video Bench | | | | | MLVU | Avg. |
|---|---|---|---|---|---|---|---|---|---|---|---|---|---|---|---|
| | | | | | Short | Medium | Long | Overall | 15 | 60 | 600 | 3600 | Overall | | |
| 2B | 100% | 34560 | 256 | - | 64.89 | 50.56 | 45.89 | 53.78 | 55.56 | 58.14 | 50.49 | 42.20 | 48.69 | 62.25 | 54.91 |
| | $2^{-3}$ | 4320 | 32 | Frame Uniform | 65.11 | 49.33 | 43.67 | 52.70 | 53.97 | 63.95 | 47.57 | 43.79 | 48.99 | 59.54 | 53.74 |
| | | | 256 | Pooling | 55.78 | 44.33 | 42.00 | 47.37 | 53.44 | 58.72 | 47.57 | 41.67 | 47.35 | 57.06 | 50.59 |
| | | | | LongVU | 65.00 | 52.44 | 47.11 | 54.85 | 56.09 | 59.88 | 48.30 | 41.67 | 48.09 | 60.46 | 54.47 |
| | | | | TS-LLaVA | 66.89 | 52.33 | 45.33 | 54.85 | 56.09 | 62.21 | 47.57 | 42.73 | 48.62 | 61.10 | 54.86 |
| | | | | DivPrune | 65.44 | 50.78 | 45.44 | 53.89 | 55.03 | 61.63 | 50.49 | 42.38 | 49.14 | 56.76 | 53.26 |
| | | | | K-means | 63.67 | 49.67 | 44.11 | 52.48 | 56.61 | 62.21 | 46.85 | 44.68 | 49.29 | 60.69 | 54.15 |
| | | | | K-medoids | 63.44 | 51.56 | 44.89 | 53.30 | 55.03 | 62.79 | 46.85 | 41.14 | 47.64 | 60.18 | 53.71 |
| | | | | SC | 66.44 | 52.44 | 47.33 | 55.41 | 56.09 | 62.21 | 48.06 | 43.62 | 49.14 | 61.43 | 55.33 |
| | | | | FLoC (Ours) | 66.11 | 52.44 | 47.00 | 55.19 | 53.44 | 60.47 | 47.57 | 44.50 | 48.77 | 62.30 | 55.42 |
| | $2^{-4}$ | 2160 | 16 | Frame Uniform | 62.44 | 47.22 | 42.44 | 50.70 | 53.97 | 60.47 | 46.85 | 43.26 | 48.09 | 56.32 | 51.70 |
| | | | 256 | Pooling | 47.56 | 39.67 | 39.22 | 42.15 | 49.21 | 51.16 | 46.36 | 41.14 | 45.18 | 52.69 | 46.67 |
| | | | | LongVU | 61.56 | 47.56 | 43.78 | 50.96 | 56.09 | 59.88 | 46.12 | 42.91 | 47.94 | 55.77 | 51.56 |
| | | | | TS-LLaVA | 64.44 | 50.56 | 43.56 | 52.85 | 56.61 | 61.05 | 47.09 | 41.67 | 47.94 | 60.23 | 51.73 |
| | | | | DivPrune | 64.44 | 48.67 | 44.44 | 52.52 | 55.03 | 58.72 | 47.09 | 40.60 | 46.97 | 55.70 | 51.73 |
| | | | | K-means | 61.56 | 47.11 | 42.11 | 50.26 | 56.09 | 61.05 | 50.73 | 42.02 | 49.14 | 59.40 | 52.93 |
| | | | | K-medoids | 62.67 | 49.00 | 41.56 | 51.07 | 52.38 | 60.47 | 47.33 | 41.31 | 47.20 | 59.22 | 52.50 |
| | | | | SC | 65.22 | 51.67 | 44.78 | 53.89 | 55.56 | 59.88 | 47.57 | 45.39 | 49.36 | 59.45 | 54.23 |
| | | | | FLoC (Ours) | 64.67 | 52.78 | 45.67 | 54.37 | 55.56 | 61.63 | 49.03 | 43.97 | 49.44 | 60.74 | 54.85 |
| | $2^{-5}$ | 1080 | 8 | Frame Uniform | 58.11 | 44.67 | 41.56 | 48.11 | 53.44 | 62.79 | 46.36 | 41.84 | 47.57 | 52.74 | 49.47 |
| | | | 256 | Pooling | 44.44 | 38.89 | 38.56 | 40.63 | 47.09 | 50.00 | 45.39 | 39.72 | 43.83 | 50.34 | 44.93 |
| | | | | LongVU | 57.78 | 43.67 | 41.89 | 47.78 | 53.44 | 59.88 | 46.36 | 43.79 | 48.02 | 52.51 | 49.44 |
| | | | | TS-LLaVA | 62.78 | 47.33 | 43.67 | 51.26 | 59.26 | 62.21 | 45.39 | 40.43 | 47.42 | 58.21 | 52.30 |
| | | | | DivPrune | 61.78 | 47.00 | 43.89 | 50.89 | 53.44 | 58.72 | 46.12 | 40.96 | 46.60 | 54.19 | 50.56 |
| | | | | K-means | 56.33 | 44.33 | 40.44 | 47.04 | 55.56 | 58.14 | 48.30 | 40.60 | 47.35 | 57.38 | 50.59 |
| | | | | K-medoids | 58.89 | 45.56 | 41.11 | 48.52 | 52.38 | 55.81 | 45.15 | 41.67 | 46.07 | 56.00 | 50.20 |
| | | | | SC | 63.00 | 49.56 | 45.00 | 52.52 | 57.67 | 56.40 | 47.33 | 42.38 | 47.87 | 58.62 | 53.00 |
| | | | | FLoC (Ours) | 64.22 | 49.00 | 45.00 | 52.74 | 57.67 | 60.47 | 48.06 | 40.96 | 48.02 | 59.31 | 53.36 |
| 7B | 100% | 34560 | 256 | - | 72.10 | 63.20 | 53.90 | 63.07 | 64.55 | 71.51 | 54.85 | 48.05 | 55.50 | 64.69 | 61.09 |
| | $2^{-3}$ | 4320 | 32 | Frame Uniform | 71.00 | 56.00 | 48.89 | 58.63 | 67.73 | 70.93 | 53.64 | 46.99 | 55.05 | 64.51 | 59.40 |
| | | | 256 | Pooling | 63.33 | 50.89 | 46.00 | 53.41 | 60.32 | 62.79 | 51.70 | 48.23 | 52.88 | 63.40 | 56.56 |
| | | | | LongVU | 71.11 | 57.67 | 47.89 | 58.89 | 68.78 | 73.26 | 53.16 | 49.47 | 56.40 | 65.01 | 60.10 |
| | | | | TS-LLaVA | 72.40 | 59.60 | 50.80 | 60.93 | 68.25 | 73.26 | 56.31 | 49.11 | 57.14 | 66.53 | 61.53 |
| | | | | DivPrune | 71.22 | 59.00 | 51.78 | 60.67 | 69.31 | 72.67 | 57.52 | 49.11 | 57.59 | 65.82 | 61.36 |
| | | | | K-means | 69.00 | 55.00 | 46.78 | 56.93 | 67.20 | 72.67 | 57.77 | 46.45 | 56.25 | 64.69 | 59.29 |
| | | | | K-medoids | 70.33 | 59.78 | 50.89 | 60.33 | 63.49 | 64.54 | 52.67 | 46.81 | 53.25 | 65.10 | 59.56 |
| | | | | SC | 71.22 | 61.00 | 51.00 | 61.07 | 67.73 | 72.67 | 58.01 | 47.87 | 56.99 | 67.36 | 61.81 |
| | | | | FLoC (Ours) | 72.00 | 60.22 | 50.44 | 60.89 | 69.84 | 72.09 | 57.04 | 50.00 | 57.82 | 67.77 | 62.16 |
| | $2^{-4}$ | 2160 | 16 | Frame Uniform | 67.22 | 53.00 | 47.22 | 55.81 | 64.55 | 70.93 | 54.37 | 46.81 | 54.75 | 61.10 | 57.22 |
| | | | 256 | Pooling | 57.78 | 48.33 | 43.78 | 49.96 | 54.50 | 60.47 | 48.54 | 45.39 | 49.59 | 60.92 | 53.49 |
| | | | | LongVU | 65.89 | 54.00 | 47.78 | 55.89 | 64.55 | 67.44 | 51.46 | 46.45 | 53.25 | 61.70 | 56.95 |
| | | | | TS-LLaVA | 70.00 | 55.70 | 48.40 | 58.04 | 67.73 | 70.35 | 54.13 | 49.82 | 56.32 | 64.69 | 59.68 |
| | | | | DivPrune | 70.67 | 57.00 | 50.33 | 59.30 | 66.14 | 72.67 | 56.80 | 47.16 | 56.10 | 63.62 | 59.67 |
| | | | | K-means | 65.78 | 52.89 | 47.22 | 55.30 | 65.61 | 70.93 | 56.55 | 46.81 | 55.57 | 62.76 | 57.88 |
| | | | | K-medoids | 69.67 | 56.89 | 50.78 | 59.11 | 61.38 | 65.70 | 51.70 | 45.92 | 52.43 | 61.43 | 57.66 |
| | | | | SC | 68.78 | 57.89 | 51.00 | 59.22 | 65.08 | 70.35 | 56.55 | 46.81 | 55.42 | 65.79 | 60.14 |
| | | | | FLoC (Ours) | 69.22 | 58.00 | 51.00 | 59.41 | 65.61 | 72.67 | 55.83 | 49.11 | 56.55 | 66.94 | 60.97 |
| | $2^{-5}$ | 1080 | 8 | Frame Uniform | 62.78 | 49.89 | 46.89 | 53.19 | 61.91 | 65.12 | 51.21 | 43.97 | 51.46 | 57.56 | 54.07 |
| | | | 256 | Pooling | 53.67 | 47.22 | 42.56 | 47.81 | 53.97 | 54.65 | 44.90 | 43.09 | 46.67 | 57.98 | 50.82 |
| | | | | LongVU | 63.70 | 50.20 | 48.70 | 54.19 | 62.96 | 65.12 | 50.97 | 44.50 | 51.76 | 57.61 | 54.52 |
| | | | | TS-LLaVA | 67.40 | 54.30 | 48.30 | 56.70 | 68.25 | 68.61 | 53.40 | 47.34 | 54.90 | 61.66 | 57.75 |
| | | | | DivPrune | 68.22 | 54.00 | 49.33 | 57.07 | 66.13 | 70.35 | 53.64 | 46.81 | 54.67 | 61.22 | 57.65 |
| | | | | K-means | 61.11 | 50.56 | 46.56 | 52.74 | 61.91 | 63.37 | 53.40 | 45.04 | 52.36 | 60.18 | 55.09 |
| | | | | K-medoids | 65.22 | 51.44 | 46.33 | 54.33 | 58.20 | 63.37 | 49.27 | 43.97 | 50.11 | 58.99 | 54.48 |
| | | | | SC | 67.89 | 53.56 | 48.33 | 56.59 | 64.55 | 69.19 | 52.18 | 46.54 | 53.70 | 63.26 | 57.85 |
| | | | | FLoC (Ours) | 69.33 | 55.56 | 48.22 | 57.70 | 64.55 | 69.77 | 54.37 | 47.34 | 54.82 | 64.23 | 58.92 |

divided into a total of seven sub-categories: Temporal Recognition (TR), Action Recognition (AR), Needle Question Answering (NQA), Ego Reasoning (ER), Plot Question Answering (PQA), Action Order (AO), and Action Count (AC).

As demonstrated in the Table 6, our proposed algorithm consistently achieved the best performance across all compression ratios for two specific tasks: **Needle Question Answering (NQA)** and **Ego Reasoning (ER)**. The **NQA** task involves inserting a relatively very short video segment, with content entirely different from the original video, into a long video sequence and then posing questions about this inserted segment. The **Ego Reasoning (ER)** task predominantly features questions about the location or state of objects that appear fleetingly in videos recorded from a first-person perspective (e.g., a user wearing a smart device while navigating daily life or performing tasks).

When conventional token compression methods are applied to such tasks, critical information pertaining to these fine details can be easily lost during the compression process. However, the empirical results robustly demonstrate that our proposed algorithm maintains its effectiveness in these

Table 6: Performance comparison on MLVU sub-tasks across different compression ratios. Our proposed method is highlighted.

| Ratio | Methods | Holistic | | Single Detail | | | Multi Detail | | Overall |
|---|---|---|---|---|---|---|---|---|---|
| | | TR | AR | NQA | ER | PQA | AO | AC | |
| $2^{-5}$ | Frame Uniform | 80.68 | 65 | 54.37 | 47.16 | 56.03 | 41.7 | 26.7 | 53.66% |
| | Pooling | 81.44 | 52 | 59.15 | 47.16 | 57.7 | **47.1** | **32.52** | 54.94% |
| | K-means | 84.85 | 62.5 | 62.82 | 52.27 | **66.79** | 46.33 | 28.16 | 58.49% |
| | K-medoids | 85.98 | 63 | 58.87 | 50.28 | 56.59 | 43.63 | 27.67 | 55.82% |
| | SC | **86.74** | 65.5 | 61.13 | 52.84 | 64.75 | 45.17 | 30.58 | 59.40% |
| | LongVU | 80.68 | 61.5 | 52.96 | 46.59 | 57.51 | 42.86 | 27.67 | 53.61% |
| | TS-LLaVA | 85.61 | 65 | 59.44 | 50.85 | 61.78 | 44.4 | 27.67 | 57.52% |
| | DivPrune | 85.17 | **69** | 68.45 | 54.55 | 60.85 | 44.79 | 24.76 | 59.43% |
| | Ours | 85.17 | 65.5 | **71.27** | **56.25** | 66.79 | 45.17 | 23.3 | **61.22%** |
| $2^{-4}$ | Frame Uniform | 81.44 | 68.5 | 57.18 | 50.57 | 61.22 | 44.4 | 32.04 | 57.20% |
| | Pooling | 82.95 | 57 | 64.79 | 48.58 | 61.41 | 48.26 | 30.1 | 57.56% |
| | K-means | 85.98 | 68.5 | 69.01 | 54.26 | 69.57 | 48.65 | **34.47** | 63.08% |
| | K-medoids | **87.12** | 63.5 | 62.82 | 52.27 | 64.01 | 44.79 | 30.1 | 59.17% |
| | SC | 88.64 | 70 | 67.04 | 54.55 | **72.17** | **50.97** | 33.01 | 64.05% |
| | LongVU | 84.47 | 66.5 | 57.18 | 50.57 | 59.37 | 44.79 | 29.61 | 56.74% |
| | TS-LLaVA | 85.61 | **73** | 65.35 | 52.56 | 66.98 | 50.19 | 28.16 | 61.52% |
| | DivPrune | 85.17 | 72 | 72.11 | 58.24 | 63.64 | 47.1 | 28.64 | 62.24% |
| | Ours | 84.79 | 68 | **74.93** | **59.09** | 70.5 | 49.42 | 30.1 | **64.54%** |
| $2^{-3}$ | Frame Uniform | 84.85 | 68 | 67.32 | 54.55 | 66.6 | 41.7 | 31.55 | 60.83% |
| | Pooling | 83.71 | 59.5 | 70.42 | 53.41 | 65.31 | 51.74 | 33.01 | 61.24% |
| | K-means | 84.85 | 73 | 73.52 | 60.51 | **73.65** | 52.9 | **44.66** | 66.59% |
| | K-medoids | **86.74** | 68 | 67.61 | 52.27 | 69.39 | 48.26 | 30.58 | 62.11% |
| | SC | **86.74** | 73.5 | 70.99 | 56.82 | 72.91 | **54.05** | 36.89 | 66.07% |
| | LongVU | **86.74** | **74.5** | 67.89 | 54.55 | 64.94 | 49.03 | 35.44 | 62.57% |
| | TS-LLaVA | 85.61 | 72 | 72.39 | 55.11 | 72.17 | 49.81 | 37.86 | 65.15% |
| | DivPrune | 85.17 | 71.5 | 74.08 | 60.8 | 68.27 | 50.58 | 33.98 | 65.00% |
| | Ours | 86.31 | 73.5 | **76.06** | **62.22** | 73.1 | 53.28 | 34.47 | **67.43%** |

challenging scenarios. Furthermore, it is evident that our algorithm's performance on the other tasks does not lag behind that of competing algorithms. This suggests that our approach not only preserves global contextual information but also minimizes the loss of crucial details.

In a subsequent subsection dedicated to qualitative result analysis, we will delve into a more specific examination of the visual tokens selected by our proposed algorithm.

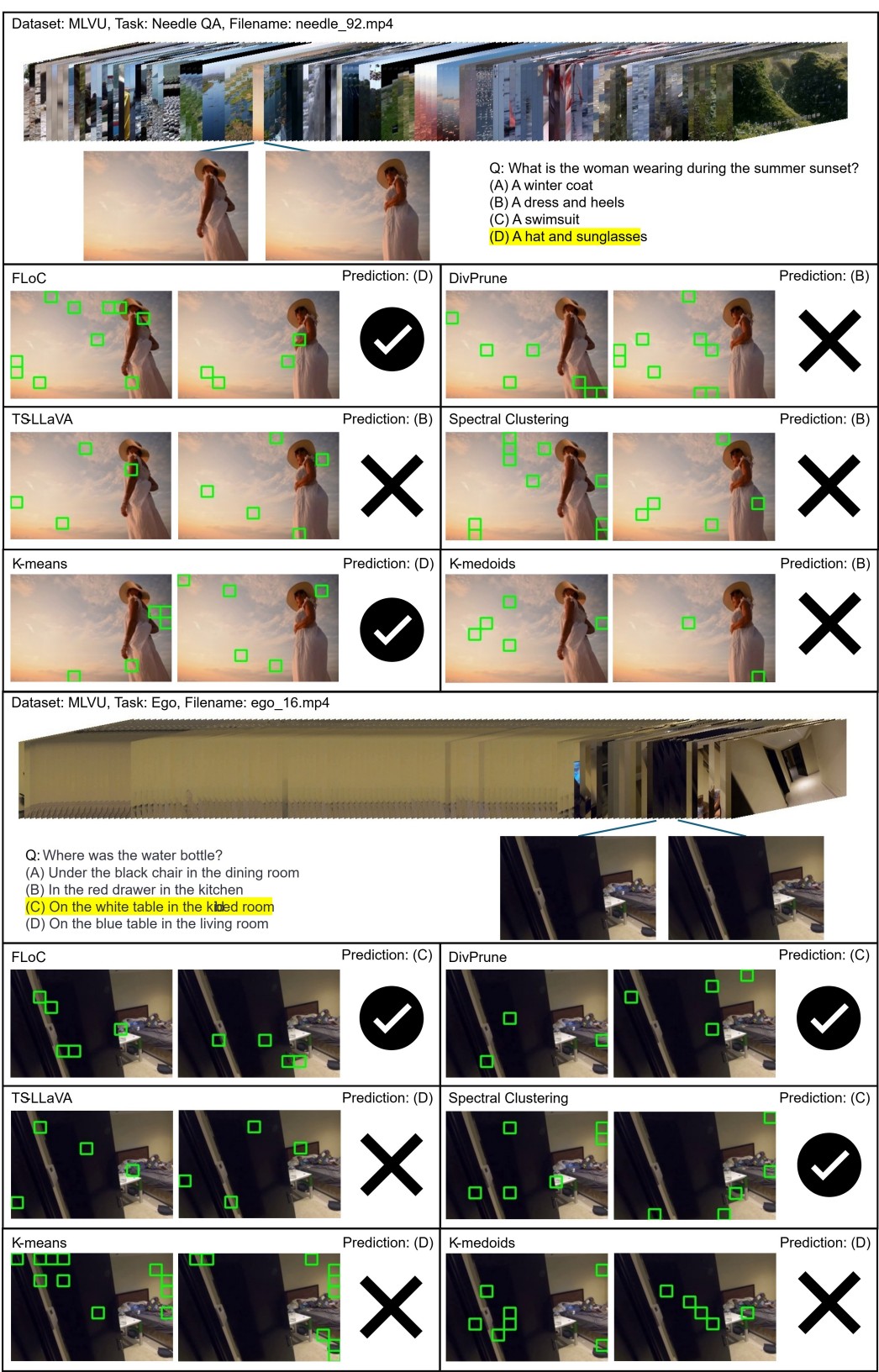

Figure 8: The first and second examples of qualitative analysis.

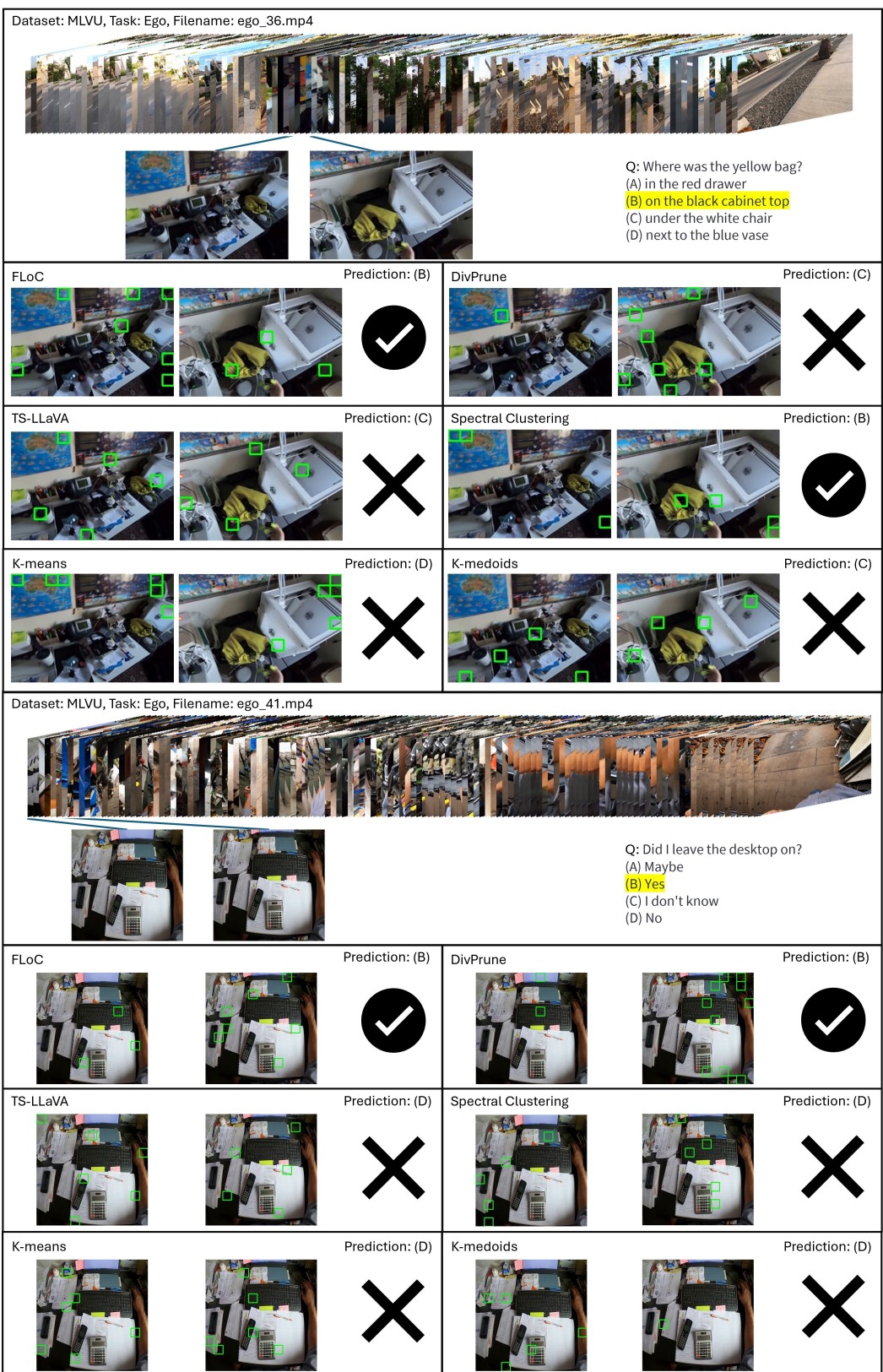

Figure 9: The third and fourth examples of qualitative analysis.

## D    QUALITATIVE RESULT ANALYSIS

To further substantiate the efficacy of our proposed visual token compression algorithm, we conducted a qualitative analysis. This analysis specifically focuses on examples from the MLVU dataset, particularly the **needle QA** and **ego reasoning** tasks, where our method demonstrated pronounced performance gains. We meticulously examined four video-question pairs, comparing the token selection and prediction outcomes of our algorithm against baseline methods. For all experiments, the compression ratio was uniformly set to $1/32$. The first three examples were processed using the Qwen2-vl 7B model, while the final example utilized the Llava Next Video Qwen 7B model.

As illustrated in Fig. 8 and Fig. 9, these tasks present a significant challenge: they require the identification of minute details within long video sequences, often spanning hundreds of frames, where the crucial information for answering the question is embedded in only a few key frames. The visual tokens selected by each compression algorithm are highlighted with green bounding boxes overlaid on their corresponding patches in the video frames.

The results compellingly demonstrate our algorithm's superior ability to pinpoint the decisive visual tokens essential for inferring the correct answer in all evaluated scenarios.

- In the **first example**, our method successfully identified patches corresponding to the woman's **sunglasses and hat**, leading to the correct answer.
- For the **second example**, the crucial visual tokens representing the **water bottle on the white table** were accurately selected.
- In the **third example**, our algorithm focused on the **yellow bag placed on the black cabinet**.
- The **fourth example** saw our method select patches depicting the **powered-on monitor**.

Consequently, our algorithm correctly answered all four questions.

In stark contrast, the baseline algorithms rarely selected the visual tokens corresponding to these critical objects. While they occasionally managed to infer the correct answer by selecting nearby or contextually related tokens, they failed in the majority of these challenging instances. This observation underscores the baselines' limitations in preserving fine-grained details under high compression.

These qualitative findings strongly suggest that our proposed algorithm can effectively retain detailed visual information, even at an extreme compression ratio such as $1/32$. This capability is paramount for tasks that demand a granular understanding of visual content within extensive video data. The ability to isolate and preserve these "needle-in-a-haystack" visual cues is a key differentiator of our approach.

## E    BASELINE IMPLEMENTATION DETAILS

This section outlines the implementation specifics and hyperparameter settings for the baseline algorithms used in our experiments.

- **K-means, K-medoids, Spectral Clustering:** For these clustering-based approaches, we utilized the scikit-learn library, employing its default parameters. For k-means and spectral clustering, after determining the clusters, the representative token for each cluster was selected as the token closest to the mean of all tokens within that cluster. Due to a significant increase in computation time with larger block sizes, the block size was set to 8 for these methods.
- **LongVU:** We implemented and utilized only the spatial token compression component of LongVU, excluding the query-based cross-attention mechanism. To ensure precise control over the compression ratio, which is not achievable with a fixed similarity threshold, we implemented an adaptive thresholding mechanism. This approach dynamically determines the appropriate threshold value to merge token pairs based on their similarity, thereby achieving the target compression ratio.
- **PruneVID:** We utilized only the first stage of the algorithm, which performs query-agnostic spatial-temporal token merging. To ensure a fair comparison, the subsequent query-aware

stage was excluded. All experiments were conducted based on the official GitHub repository provided by the authors.

- **DyCoke:** we adopted the query-agnostic compression component corresponding to Stage 1, specifically the visual token temporal merging module. The implementation was based on the official GitHub repository provided by the authors.

- **TS-LLaVA:** TS-LLaVA originally combines two strategies: creating thumbnails from raw frames and uniformly sampling visual tokens. However, in our experiments with the selected benchmark datasets and backbone LLMs, incorporating the thumbnail generation aspect led to a degradation in performance. Consequently, we only included the uniform token sampling component of TS-LLaVA in our baseline comparisons.

- **DivPrune:** Due to code compatibility issues with the officially provided GitHub repository, we re-implemented DivPrune based on the pseudo-code presented in its original publication. The algorithm was straightforward to implement from the provided pseudo-code. For our experiments, the block size for DivPrune was set to 32.

- **STTM and LLaVA Scissor:** We used the official implementations provided by the authors for all benchmarks. Hyperparameters were kept at their default settings, while threshold values were adjusted to achieve the desired compression ratio.

- **FastVID:** We conducted experiments based on the official GitHub repository provided by the authors. Among the models we tested, implementation was available only for the Qwen2.5-VL model; therefore, experiments were limited to this model. We varied only the retention ratio while keeping all other hyperparameters at their default values.

## F  T-SNE VISUALIZATION OF TOKEN DISTRIBUTIONS

While a t-SNE visualization of the selected token distributions was included in the originally submitted manuscript, space constraints necessitated the use of smaller images. For enhanced clarity and easier inspection, we have attached larger versions of these visualizations in Fig 10.

These visualizations demonstrate that the tokens selected by our proposed method more uniformly cover the entire t-SNE distribution compared to those selected by other baseline approaches. Notably, while the DivPrune method also aims to select diverse tokens based on a min-max distance criterion, its chosen tokens do not achieve the same level of even coverage across the entire distribution as observed with our algorithm. This suggests our method is more effective at capturing a comprehensive and representative set of visual features.

## G  PROFILING OF COMPUTATIONAL AND MEMORY FOOTPRINT

| Method | Inference (s) | Compression (s) | Total (s) | GFLOPS | VRAM (GB) |
|--------|---------------|-----------------|-----------|--------|-----------|
| Full | 3.22 | – | 2.69 | – | 27.33 |
| FLoC | | 0.99 | 3.00 | 318 | 17.96 |
| DyCoke | | 0.13 | 1.83 | 0.46 | 17.96 |
| PruneVID | | 0.19 | 1.89 | 1.69 | 17.96 |
| STTM | | 0.07 | 1.77 | 0.17 | 17.96 |
| Scissor | | 1.24 | 2.94 | 783 | 17.96 |
| FastVID | 1.70 | 0.44 | 2.14 | 6 | 17.96 |
| DivPrune | | 2.37 | 4.07 | 317 | 17.96 |
| LongVU | | 0.38 | 2.08 | 317 | 17.96 |
| kmeans | | 12.31 | 14.01 | 82 | 17.96 |
| kmedoids | | 2.41 | 4.11 | 82 | 17.96 |
| spectral | | 8.12 | 9.82 | 82 | 17.96 |

Table 7: Performance comparison of different methods

To compare the resource consumption and speed of our proposed algorithm against baseline methods, we measured LLM inference time, compression time, FLOPs, and peak VRAM usage. All experiments were conducted using the Qwen2.5-VL 7B model on an NVIDIA H100 GPU, with a compres-

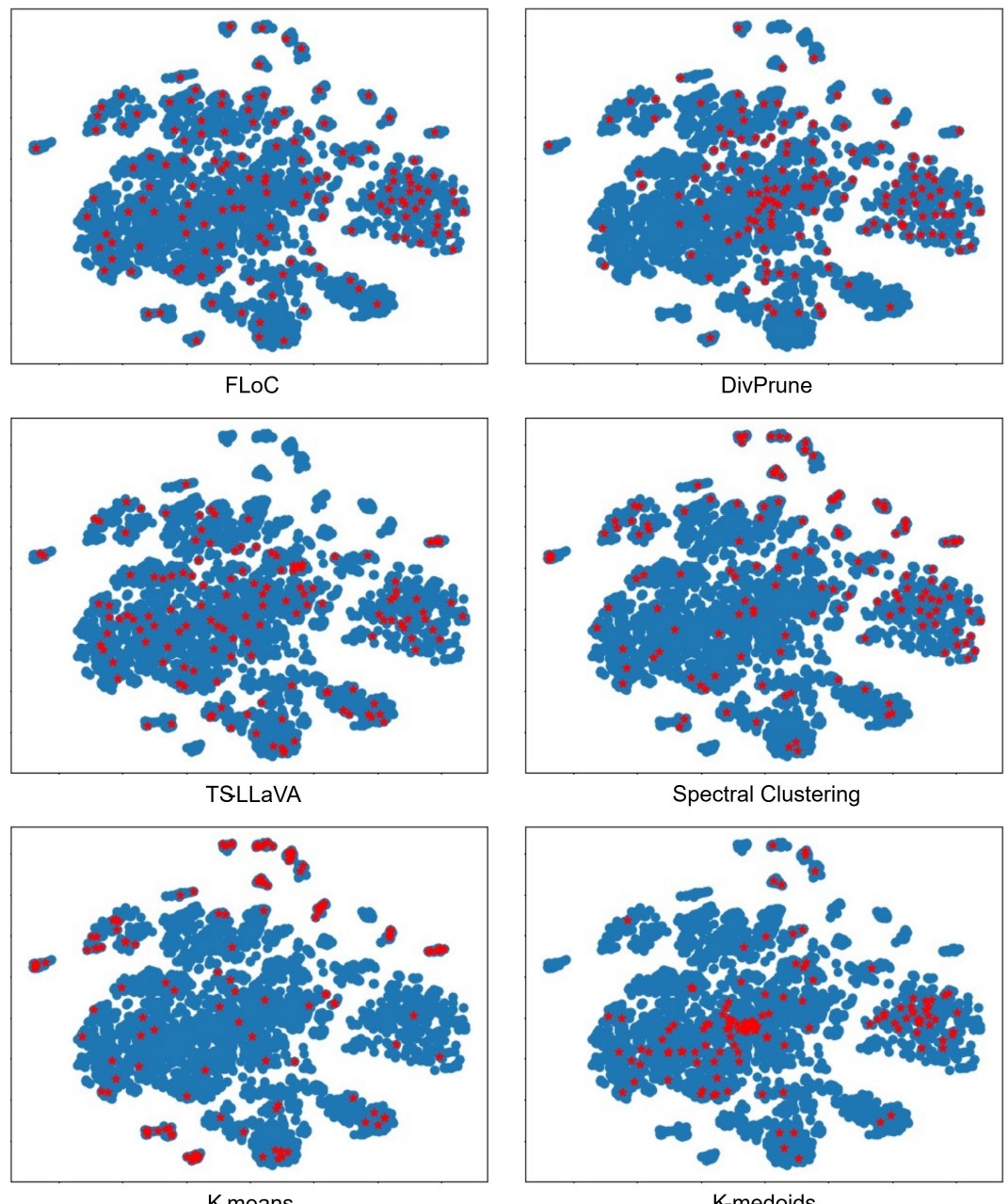

Figure 10: T-SNE plots for proposed and other visual token compression algorithms.

sion ratio of 12.5%. The reported times correspond to end-to-end processing of 784 frames, each containing 60 visual tokens. FLOPs were estimated by accumulating the actual operation counts during execution, with approximations for modularized components (which may introduce minor inaccuracies). Peak VRAM usage was recorded using PyTorch's `torch.cuda.max_memory_allocated()` function.

Overall, graph-based methods tend to achieve faster compression and require fewer operations, but—as shown in previous experiments—they exhibit inferior performance. In contrast, clustering-based methods incur significantly higher computational costs, resulting in slower processing speeds. Our proposed algorithm performs compression in less time than the LLM inference step, demonstrating practical efficiency. However, it exhibits relatively high FLOPs, most of which are attributed to pairwise similarity computations.

## H   ABLATION STUDY ON SIMILARITY METRICS IN FACILITY LOCATION

To investigate the impact of the similarity measure on the facility location function, we conducted an ablation study comparing our default **Cosine Similarity** with **Euclidean Distance**. Since the facility location function requires a similarity matrix, we converted the Euclidean distance into a similarity measure using a Gaussian kernel:

$$S_{euc}(x, y) = \exp\left(-\frac{\|x - y\|^2}{2\sigma^2}\right)$$

where $\sigma$ is set to the median of all pairwise Euclidean distances within the set, following the standard median heuristic. We performed experiments using the **InternVL3-8B** model across different compression ratios (1/8, 1/16, and 1/32).

Table 8: Performance with different distance metrics.

| $T$ | Metric | VideoMME | MLVU | LVB | EgoSchema | Average | Difference |
|---|---|---|---|---|---|---|---|
| $2^{-3}$ | Cosine | 64.93 | 71.57 | 56.69 | 69.40 | 65.65 | +0.55 |
| | Euclidean | 64.26 | 72.26 | 58.49 | 69.80 | 66.20 | |
| $2^{-4}$ | Cosine | 63.41 | 69.09 | 56.47 | 66.20 | 63.79 | -0.27 |
| | Euclidean | 62.74 | 69.50 | 55.42 | 66.40 | 63.52 | |
| $2^{-5}$ | Cosine | 60.81 | 66.93 | 54.23 | 63.80 | 61.44 | -0.97 |
| | Euclidean | 59.59 | 65.59 | 53.70 | 63.00 | 60.47 | |

The results are summarized in Table 8. We observed that while the Euclidean-based metric showed a slight advantage (+0.55 accuracy) at a low compression ratio (1/8), **Cosine Similarity consistently outperformed Euclidean similarity as the compression ratio increased**. Specifically, Cosine similarity achieved higher accuracy at 1/16 (+0.27) and 1/32 (+0.97) ratios.

This suggests that while Euclidean distance captures fine-grained magnitude differences useful when retaining many tokens, **Cosine similarity is more robust for abstract feature space coverage**, particularly in high-compression regimes where capturing the dominant semantic directions is crucial. Based on these findings, we adopted Cosine similarity as the default metric to ensure consistent performance across varying degrees of compression.

## I   LIMITATIONS AND FUTURE DIRECTIONS

A key limitation of the proposed **FLoC** algorithm lies in the empirical determination of its sole hyperparameter: the block length ($T$). The choice of $T$ involves a critical trade-off that can impact both performance and computational efficiency.

- **Longer block lengths** allow the algorithm to consider representativeness and diversity over a more extended temporal context. This can be advantageous for capturing the nuances of slowly evolving scenes. However, it also leads to a proportional increase in computational overhead during the token selection process.

- **Shorter block lengths** reduce the computational cost. However, they can introduce a risk of inter-block redundancy. For example, if a long, static scene is segmented into multiple short blocks, the algorithm might select very similar (or even identical) tokens from each block. This diminishes the diversity of the final selected set, as redundancy is only minimized within each block, not across them.

This trade-off implies that the optimal setting for the block length is content-dependent. For instance, a static video (e.g., a lecture) might benefit from a longer block length, whereas a highly dynamic video with frequent cuts may be better served by a shorter one.

A promising direction for future work is to develop a method for automatically determining the block length. One could, for example, employ a pre-processing step using a scene detection algorithm. By aligning block boundaries with detected scene changes, the algorithm could dynamically adapt the block length to the video's temporal structure. This would not only make the framework more robust

but could also further enhance performance by ensuring that each block represents a semantically coherent segment, thereby mitigating inter-block redundancy and improving the quality of the selected tokens.

