# OpenReview forum: "FLoC: Facility Location-Based Efficient Visual Token Compression for Long Video Understanding"
_ICLR.cc/2026/Conference — ICLR 2026 Poster_

### Official Review · Reviewer_kqKm · 2025-10-30

**Soundness:** 3
**Presentation:** 2
**Contribution:** 2
**Rating:** 6
**Confidence:** 4

**Summary:**

This paper introduces FLoC, a training-free and model-agnostic visual token compression framework for long video understanding. By using a facility location function and a lazy greedy algorithm, FLoC efficiently selects a compact and representative subset of visual tokens, significantly reducing token volume while maintaining near-optimal performance. Extensive experiments show that FLoC outperforms recent compression methods on large-scale benchmarks in both effectiveness and processing speed.

**Strengths:**

1. The objective of the paper is well-motivated and clearly stated.

2. The proposed method achieves state-of-the-art performance and efficiency in nearly all cases.

3. The authors conduct comprehensive experiments across multiple benchmarks and baseline MLLMs.

**Weaknesses:**

1. The authors should evaluate throughput, inference speed, and GPU memory usage to further highlight the efficiency of the proposed method compared to existing approaches.

2. In the comparison experiments, it is surprising that PruneVid performs worse than random selection, given that PruneVid employs both clustering and a query-aware selection algorithm. Some discussion or analysis of this result would be helpful.

**Questions:**

1. As shown in Table 3, basic clustering methods already achieve comparable performance. Does this suggest that sophisticated or complex designs may not be necessary?

---

> ### Author Response · Authors · 2025-11-21
>
> ### **(W1) Evaluation of Throughput, Speed, and Memory (Appendix G)**
>
> We have addressed this by providing a detailed efficiency profile in Appendix G. **Table 7** now reports the inference speed, total latency, GFLOPS, and peak GPU memory (VRAM) usage for FLoC and all baseline methods. These results further highlight the computational efficiency of our proposed framework compared to existing approaches.
>
> ---
>
> ### **(W2) Analysis of PruneVid Performance (Query-Agnostic Setting & Merging Artifacts)**
>
> We thank the reviewer for pointing this out. We would like to clarify the experimental setting and provide an analysis of why **PruneVid underperforms in our comparison**.
>
> #### 1. Experimental Setting (Fair Comparison)
> As detailed in **Appendix E**, for a fair comparison with other generic token compression methods, we evaluated **only Stage 1 (Spatial-Temporal Token Merging)** of PruneVid, which operates in a query-agnostic manner. The query-aware selection stage was excluded to align with the query-agnostic nature of the baselines (e.g., Random, K-Means, FLoC).
>
> #### 2. Analysis of Poor Performance
> The performance degradation of PruneVid (often falling below Random selection) stems from inherent limitations in its graph-based merging mechanism:
>
> - **Weak Connection Problem:** PruneVid merges tokens based on a local similarity threshold. This often leads to the *weak connection phenomenon*, where semantically unrelated and distant tokens are chained together through a series of locally weak similar links (slightly over the threshold value).
> - **Semantic Distortion via Averaging:** Unlike clustering methods that "select" a representative token (e.g., medoids), PruneVid generates a new centroid vector by computing the mean of the merged component. When unrelated tokens (e.g., a foreground object and a background patch) are merged due to weak connections, averaging their features creates a new embedding that represents neither, effectively acting as noise.
>
> This combination of **incorrect grouping (weak connection)** and **feature distortion (averaging)** explains why PruneVid yields lower performance compared to Random selection, which at least preserves the original semantic integrity of the selected tokens.
>
> ---
>
> ### **(Q1) Why Sophisticated Designs are Necessary (Hardware Feasibility)**
>
> We agree that basic clustering methods achieve comparable accuracy, raising the valid question of whether specialized designs are necessary. However, we argue that **"conceptual simplicity" does not imply "computational or hardware simplicity."** FLoC is necessary precisely because it addresses the deployment bottlenecks of clustering methods.
>
> ---
>
> #### *Hardware Feasibility (Ops Complexity)*
> - While clustering concepts are simple, their implementation often requires **complex linear algebra operations** (e.g., eigen-decomposition in spectral clustering).
> - These operations are computationally expensive and difficult to implement efficiently on **resource-constrained edge devices** (e.g., NPUs/DSPs).
>
> ---
>
> #### *Implementation Simplicity*
> - In contrast, **FLoC relies almost exclusively on primitive operations—specifically dot products and simple subtraction for marginal gain updates**.
> - This makes FLoC significantly more **hardware-friendly and easier to deploy on on-device AI accelerators** compared to "simple" clustering methods.
>
> ---
>
> #### *Latency*
> - As shown in **Table 3**, clustering methods are **orders of magnitude slower** due to their iterative nature.
> - **FLoC’s greedy approach guarantees a fixed, low latency**, which is a prerequisite for real-time long video understanding.
>
> ---
>
> Therefore, **specialized designs like FLoC are essential not just for accuracy, but to bridge the gap between theoretical performance and practical, on-device feasibility**.
>
> ---
>
> **We sincerely thank the reviewer for raising this important point, which allowed us to clarify the practical motivation behind our design choices.**

---

> > ### Comment · Reviewer_kqKm · 2025-11-24
> >
> > Thanks for the rebuttal. My concerns have been addressed.

---

> ### Author Response · Authors · 2025-11-24
>
> We sincerely thank you for the prompt feedback and are glad to hear that our rebuttal—particularly the efficiency profiling and the PruneVid analysis—has successfully addressed your concerns.
>
> We believe that incorporating your suggestions has significantly strengthened the empirical rigor of our work.
>
> We remain fully available for any further discussion should there be any remaining details to clarify before the end of the rebuttal.
>
> Best regards,
>
> The Authors

---

### Official Review · Reviewer_nG1P · 2025-10-30

**Soundness:** 3
**Presentation:** 3
**Contribution:** 3
**Rating:** 6
**Confidence:** 4

**Summary:**

This paper proposes a facility-location-based efficient visual-token compression framework named FLoC for long video understanding. Specifically, FLoC includes two core modules: a Visual Token Selection Module and a Compressed Embedding Module, optimized through a lazy-greedy algorithm. The Visual Token Selection Module tokenizes videos into visual tokens, leverages the submodular facility-location function to measure token representativeness and diversity, and compresses them into compact embeddings. The Compressed Embedding Module concatenates the selected token embeddings with text prompts and feeds them into video-LMMs for downstream tasks. The entire process is guided by the greedy selection strategy, ensuring semantic fidelity and efficient compression without fully decoding the video.

The experimental evaluation in this paper assessed the performance of the proposed FLoC across three compression ratios and three video understanding benchmarks, comparing it with state-of-the-art baselines. The results indicate that the proposed FLoC achieves superior accuracy while reducing sequence length, thereby providing computational and memory efficiency gains.

**Strengths:**

- The compression process does not rely on specific model architectures, nor is it tailored to particular queries or tasks; a single compression can support multiple downstream applications.
- A visualization video of the lazy-greedy algorithm is provided in the supplementary material, making it more persuasive.
- The experiments are sufficient, with the method’s performance verified on multiple models and benchmarks.

**Weaknesses:**

- The method performs diversity selection within individual blocks, with no information interaction between blocks. Therefore, the choice of hyper-parameter T is crucial: a T that is too large may cause redundant computation, while a T that is too small may lead to redundant similar tokens across blocks. In Tab. 4, the optimal T differs under different compression rates, making it difficult to select an appropriate T across settings with different models, compression rates, and benchmarks.
- FLoC aims for “global coverage,” which may cause extremely sparse but important tokens (e.g., a key object appearing in only one frame) to contribute very little to the total sum and risk being discarded, thereby affecting performance on needle-in-a-haystack tasks.
- There is no comparison with graph-based video LLM compression methods such as FastVID or LLaVA-Scissor.
- Only computation times of FLoC and traditional algorithms are compared; FLOPs and actual benchmark latency versus other compression methods are not reported.

**Questions:**

Please refer to Weaknesses.

---

> ### Author Response · Authors · 2025-11-21
>
> ### **Extensive Experiments on Hyper-parameter T Across Datasets (Figure 6)**
>
> We sincerely appreciate the reviewer’s valuable feedback regarding the sensitivity of the hyper-parameter **T** and the potential difficulty in its selection. Motivated by your comment, we conducted **extensive additional experiments across diverse datasets** (Video-MME, MLVU, LongVideoBench) and varying compression ratios (1/8, 1/16, 1/32) to rigorously analyze the impact of T. The results of this comprehensive study are newly visualized in **Figure 6** of the revised manuscript.
>
> Our findings from Figure 6 address the raised concerns as follows:
>
> ---
>
> #### **Performance Saturation**
> - While the absolute peak varies slightly across settings, the performance curves consistently **saturate as T increases (typically T ≥ 32)**.
> - This indicates a broad *“safety zone”* where performance remains near-optimal, implying that **precise tuning for every specific setting is not required**.
>
> ---
>
> #### **Robust Default**
> - Based on these extensive empirical results, we identified that a **fixed value (e.g., T=32)** serves as a robust default that consistently delivers high performance across different benchmarks and compression rates.
>
> ---
>
> #### **Efficiency Regarding “Redundant Computation”**
> - Addressing the concern that a large T might lead to redundant computation, our **lazy greedy implementation ensures that increasing T incurs negligible latency overhead**.
> - As discussed in Section 4.5.3, this efficiency allows us to use a sufficiently large T (to guarantee coverage) **without the computational penalty typically associated with larger block sizes in other methods**.
>
> ---
>
> We believe these additional experimental results, prompted by your insight, clearly demonstrate that **our method is practically robust and does not require burdensome hyper-parameter tuning**.
>
> ---
>
> **We sincerely thank the reviewer for this insightful comment, which helped us strengthen the empirical validation of our approach.**

---

> ### Author Response · Authors · 2025-11-21
>
> ### **Preservation of Sparse Tokens and Needle-in-a-Haystack Performance**
>
> We appreciate the reviewer’s thoughtful concern regarding the potential risk of discarding sparse but critical tokens ("needles") when optimizing for global coverage. We clarify that our **Greedy Facility Location mechanism inherently prioritizes these sparse tokens in later iterations based on marginal gain**, which is strongly supported by our quantitative, qualitative, and visual evidence.
>
> ---
>
> #### 1. Mechanism: Marginal Gain and Diversity
> - While the objective is global coverage, our iterative greedy algorithm selects tokens based on the increase in coverage (**marginal gain**).
> - Once the dominant/redundant patterns (backgrounds) are covered in early iterations, the marginal gain of selecting similar tokens drops to near zero.
> - Consequently, the algorithm naturally prioritizes **diverse, uncovered tokens—even if they appear in only a single frame—because they offer the highest remaining gain by filling the coverage gaps**.
>
> ---
>
> #### 2. Quantitative Evidence (Needle-in-a-Haystack Tasks)
> This theoretical advantage translates directly into performance. As shown in **Table 6 (Appendix C)** on the MLVU benchmark, our method excels in tasks specifically designed to test fine-grained detail retention:
>
> - **Needle QA (NQA):** This task evaluates the ability to answer questions about a very short, unrelated video segment inserted into a long sequence (similar to the needle-in-a-haystack task).
>   **FLoC consistently outperforms the strongest baseline (DivPrune) across all compression ratios** (e.g., 71.27% vs. 68.45% at 1/32 compression).
> - **Ego Reasoning (ER):** This task requires reasoning about fleeting objects (e.g., location or state) that appear momentarily in first-person perspective videos.
>   **FLoC achieved significantly improved performance across all compression ratios.**
>
> ---
>
> #### 3. Qualitative Evidence (Visual Examples)
> For more concrete evidence, we refer the reviewer to the qualitative examples in **Appendix D (Figures 8 & 9)**. These figures visualize instances where the camera moves quickly or focuses transiently on small objects (e.g., sunglasses, a water bottle, or a yellow bag).
> Despite being "sparse" in the temporal sequence, **our algorithm successfully selects the tokens corresponding to these small, fleeting objects**, enabling the model to answer questions correctly where baselines fail.
>
> ---
>
> #### 4. Visualization (t-SNE Plot)
> Finally, the **t-SNE visualization (Figure 10 in Appendix F)** further corroborates this behavior. As observed in the plot, the tokens selected by FLoC (red stars) are not clustered solely in dense regions but are also widely distributed across the sparse, isolated regions of the feature space.
> This visually confirms that **our method effectively captures outliers and diverse tokens that contribute unique information to the global coverage**, rather than ignoring them.
>
> ---

---

> ### Author Response · Authors · 2025-11-21
>
> ### **Comparison with Graph-based Methods (FastVID, LLaVA-Scissor, STTM)**
>
> We have addressed this by incorporating **FastVID**, **LLaVA-Scissor**, and **STTM** into our baselines. Furthermore, we expanded our evaluation from **3 to 6 benchmarks** (adding LVBench, NextQA, and EgoSchema).
>
> As shown in the updated **Table 1**, our proposed **FLoC consistently outperforms these graph-based methods across all datasets and compression settings**. We also provided these algorithms' latency vs performance analysis in **Figure 1**.
>
> All three algorithms follow a **graph-based merging paradigm**, but their mechanisms differ:
>
> ---
>
> #### **STTM**
> - STTM reflects a hierarchical tree structure of video frames.
> - It divides spatio-temporally high-resolution tokens into grid regions and merges tokens within each grid if their similarity exceeds a threshold.
> - This process is repeated across multiple hierarchical levels.
> - **Limitation:** Because it relies on threshold-based merging, STTM suffers from the **weak connection problem** at high compression ratios, leading to significant performance degradation.
> - Weak connection problem: Distinct tokens (e.g., a small “needle” object vs. background) are irreversibly merged if they share minor similarities above a threshold, causing performance to drop sharply at low compression ratios.
>
> ---
>
> #### **LLaVA-Scissor**
> - This method merges tokens that are spatio-temporally adjacent when their similarity exceeds a threshold, forming connected components that are then merged.
> - **Limitation:** Similar to STTM, threshold-based merging causes weak connections at high compression ratios, resulting in poor performance.
>
> ---
>
> #### **FastVID**
> - Unlike the two methods above, FastVID uses **density-based token merging** rather than threshold-based merging.
> - While it performs better than STTM and LLaVA-Scissor, it introduces numerous additional parameters and heuristics.
> - **Limitation:** Despite these enhancements, its performance still falls short of **FLoC**.
>
> ---
>
> **In contrast, FLoC avoids these issues by maximizing coverage of selcted tokens based on the Facility Location objective, ensuring robust performance even under strong compression budgets.**
>
> ---
>
> ### **Report on FLOPs and Latency (Appendix G)**
>
> We have addressed this by providing a **detailed computational profile in Appendix G**. **Table 7 now reports the FLOPs, compression latency, and total inference time for all evaluated algorithms**, including the newly added baselines.
>
> ---
>
> **We sincerely thank the reviewer for these helpful suggestions, which allowed us to strengthen both the experimental rigor and the clarity of our manuscript.**

---

> > ### Comment · Reviewer_nG1P · 2025-11-28
> >
> > I have read the authors' detailed response and the revised manuscript. Most of my concerns have been settled. I appreciate the authors' effort in extending more baseline methods and benchmarks. I decide to maintain my score as 6.

---

> ### Author Response · Authors · 2025-11-28
>
> We sincerely thank you for reviewing our rebuttal and revised manuscript. We are glad to hear that our efforts to expand the baselines and benchmarks have successfully addressed most of your concerns.
>
> We believe that incorporating your suggestions—specifically the comparison with graph-based methods and the addition of diverse benchmarks—has significantly strengthened the empirical rigor and completeness of our paper.
>
> We remain fully available for any further discussion should there be any remaining details to clarify before the end of the rebuttal.
>
> Best regards,
>
> The authors

---

### Official Review · Reviewer_9yVb · 2025-11-01

**Soundness:** 3
**Presentation:** 3
**Contribution:** 3
**Rating:** 6
**Confidence:** 3

**Summary:**

This paper proposes FLoC (Facility Location-Based Efficient Visual Token Compression), a novel framework designed to tackle the scalability limits of video-Large Multimodal Models (LMMs) when processing long video sequences. The core challenge addressed is the overwhelming volume of visual tokens generated from extended videos, which makes end-to-end processing computationally infeasible given the limited context lengths of most LLM-based architectures.

**Strengths:**

1. Submodular Optimization via Facility Location: FLoC is the first visual token compression algorithm based on the facility location function and submodular optimization for long video understanding. This interprets token selection as maximizing a utility (or coverage) function that rewards tokens for preserving the essential information and diversity of the entire visual token set within a strict budget constraint (K).
2. Targeting Rare Information Loss: The facility location framework is specifically motivated to overcome a key limitation of clustering methods: their tendency to fail in capturing rare but important tokens (e.g., small objects like car keys) because they focus on densely populated regions in the feature space. FLoC explicitly optimizes for both representativeness and diversity simultaneously.
3. Lazy Greedy Efficiency: The implementation utilizes the lazy greedy algorithm to efficiently approximate the optimal NP-hard solution. This approach significantly reduces computational overhead by postponing the update of marginal gains until necessary, offering a novel and practical way to handle massive token sets.

**Weaknesses:**

1. The primary operational weakness of FLoC is the reliance on the empirical determination of a single hyperparameter: the block length ($T$). The paper explicitly states that the choice of T involves a critical trade-off that impacts both performance and computational efficiency. The optimal setting for the block length is acknowledged to be content-dependent; for instance, a static video (e.g., a lecture) benefits from a longer block length, while a highly dynamic video requires a shorter one.
2.  FLoC is designed as a query-agnostic and model-agnostic solution. While this offers advantages in memory efficiency (performing compression once) and flexibility, it fundamentally limits the model's performance in real-time, query-specific tasks or dynamic interactive environments. Query-aware compression methods, which FLoC contrasts itself against, "can effectively reduce the search space by focusing on what is deemed important" to the user.
3. FLoC utilizes the lazy greedy algorithm to approximate the maximization of the facility location function, which is NP-hard. While this is significantly faster than naive greedy and traditional clustering methods (e.g., 10× faster than K-Means in compression time), the complexity remains dependent on both the initial number of tokens ($n$) and the budget ($K$), scaled as $O(n⋅K)$.
4. The facility location function relies on the cosine similarity between token embeddings ($sim(v,u)$) as its similarity measure. Cosine similarity effectively measures angular distance, focusing on the direction of feature vectors, which is suitable for abstract feature space coverage. However, it may not perfectly capture complex perceptual differences or contextual relationships that could be better represented by other distance metrics.

**Questions:**

See weakness above, please.

---

> ### Author Response · Authors · 2025-11-21
>
> ### **(W1) On the Robustness of Hyperparameter T**
>
> We appreciate the reviewer’s insightful comment regarding the operational trade-offs of the block length (T). While we acknowledge in Appendix H that an adaptive T could theoretically optimize efficiency further, **empirical evidence suggests that FLoC is highly robust to the choice of T**, minimizing the need for per-video tuning in practice.
>
> #### 1. Performance Stability (Figure 6)
>
> - During the rebuttal period, we extended our evaluation of performance with respect to block length T—originally conducted on a single dataset—to three diverse datasets. The updated results are presented in Figure 6.
> - As illustrated in Figure 6, our method exhibits consistent performance across a wide range of block lengths once **T exceeds a small threshold (approx. T ≥ 4)**.
> - The performance curve effectively saturates and remains stable for larger T values (e.g., from 4 up to 256).
> - This trend holds true across diverse benchmarks, including **Video-MME** (varied content), **MLVU** (long-form), and **LongVideoBench** (highly dynamic).
> - This indicates that a fixed, sufficiently large default value (e.g., T=32) is *safe and effective* for both static and dynamic videos, negating the operational burden of tuning T for each input.
>
> #### 2. Efficiency of Larger T via Lazy Greedy (Table 3)
>
> - The reviewer correctly notes that static videos benefit from longer block lengths to reduce redundancy. A concern might be that a large fixed T is computationally expensive.
> - However, thanks to our **Lazy Greedy implementation**, the computational overhead grows very slowly as T increases.
> - Table 3 shows that increasing T from 2 to 32 results in only a marginal increase in processing time compared to clustering baselines (which scale poorly).
> - Therefore, setting a large fixed T allows us to capture global context (crucial for static videos) **without incurring prohibitive costs** usually associated with large-window processing.
>
> #### 3. Operational Simplicity
>
> While adaptive block sizing is a promising future direction, the current fixed-T approach offers a significant operational advantage: it remains **training-free and strictly plug-and-play**, requiring no auxiliary networks (e.g., scene detectors) or complex pre-processing steps. Given the empirical robustness shown above, we believe the fixed parameter strikes an optimal balance between **performance, efficiency, and ease of deployment**.
>
>
>
> ---
>
> **We sincerely thank the reviewer for these insightful comments, which helped us clarify and strengthen the practical contributions of our work.**

---

> ### Author Response · Authors · 2025-11-21
>
> ### **(W2) On the Practicality and Real-time Efficiency vs. Query-Aware Methods**
>
> We appreciate the reviewer’s comment regarding the theoretical benefits of query-aware compression. While we agree that query-aware methods can reduce the search space, we argue that **FLoC offers superior operational advantages in terms of deployment feasibility and real-time multi-turn interaction efficiency**, which are critical for practical Video-LMM applications.
>
> We address the reviewer’s concern with two key arguments:
>
> ---
> #### 1. Engineering Bottlenecks of Query-Aware Methods
>
> - Implementing query-aware compression typically requires modifying the internal mechanisms of the LLM (e.g., accessing intermediate attention maps or retraining projectors).
> - **Deployment Friction:** In real-world deployment, LLMs are served via optimized inference engines (e.g., vLLM, TensorRT-LLM) or accessed as black-box APIs. Query-aware methods that alter model internals are often incompatible with these optimized pipelines.
> - **Universal Compatibility:** In contrast, FLoC operates as a strictly model-agnostic pre-processor. It works instantly with any downstream Video-LMM without requiring architecture-specific adaptations, ensuring *future-proof deployability*.
> ---
> #### 2. Real-time Multi-turn QA via KV Cache Reuse
> - The reviewer mentioned "dynamic interactive environments." In such scenarios (e.g., a user asking a series of questions about a long video), memory efficiency and response latency are paramount.
> - **Query-Aware Limitation (Cache Invalidation):** Since query-aware methods dynamically select different visual tokens for every new query, the visual context changes per turn. This forces the system to invalidate the Key-Value (KV) cache and re-compute the visual features (prefill phase) for every single question, introducing significant latency and computational redundancy.
> - **FLoC Advantage (KV Cache Reuse):** Because FLoC selects a representative subset once, the visual tokens remain static across all queries. This allows the system to compute the Visual KV Cache only once and reuse it for all subsequent turns.
> - **Impact:** Consequently, answering follow-up questions becomes virtually instantaneous, as the computational cost is reduced to processing only the short text tokens. This capability enables users to receive answers to multiple questions in rapid succession, making FLoC a fundamentally more suitable solution for real-time, multi-turn video understanding.
> ---
> ### Conclusion
>
> While Query-Aware methods theoretically optimize for a single specific query, they incur prohibitive costs in deployment complexity and multi-turn latency. **FLoC provides a practical Pareto-optimal solution:** it ensures broad compatibility and leverages KV cache reuse to deliver unmatched speed and memory efficiency in interactive environments.
>
> ---
>
> **We sincerely thank the reviewer for these insightful comments, which helped us clarify and strengthen the practical contributions of our work.**

---

> ### Author Response · Authors · 2025-11-21
>
> ### **(W3) On the Computational Complexity and Potential for Further Acceleration**
> We appreciate the reviewer’s insightful comment on the $O(nK)$ complexity of the greedy algorithm. While our current Lazy Greedy implementation is significantly faster than the inference process (as shown in Table 3), we acknowledge that for extremely large-scale scenarios, further optimization beyond linear complexity relative to $K$ may be desirable.
>
> We would like to highlight that **FLoC is a modular framework based on Submodular Maximization**, which allows us to seamlessly adopt faster optimization algorithms to address this exact scalability concern without altering the core objective.
>
> **Path to Scalability via Stochastic Greedy**
> As the reviewer correctly implied, standard greedy algorithms depend linearly on the budget $K$. However, a key advantage of our principled formulation is the existence of **Stochastic Greedy algorithms** (Mirzasoleiman et al., 2015).
> * **Mechanism:** Instead of evaluating the marginal gain for all $n$ tokens in every iteration, Stochastic Greedy evaluates only a random subset of size $R = \frac{n}{K} \log(1/\epsilon)$.
> * **Complexity Reduction:** This significantly reduces the complexity from $O(nK)$ to **$O(n \log \frac{1}{\epsilon})$**. This effectively decouples the runtime dependency on the budget $K$ for the search space traversal.
> * **Theoretical Guarantee:** Crucially, this acceleration retains a provable approximation guarantee (e.g., $1-1/e - \epsilon$), ensuring that efficiency gains do not come at the cost of representativeness or diversity.
>
> **Conclusion**
>
> While we utilized Lazy Greedy in this work as it provided an optimal balance of speed and performance for current benchmarks, the FLoC framework is inherently "future-proof." It allows for the ready integration of stochastic optimization techniques to handle massive token sets with minimal performance loss, fundamentally addressing the theoretical scalability concern.
>
> ---
>
> ### **(W4) Applying Another Distance Metric**
>
> We thank the reviewer for the constructive suggestion regarding the choice of distance metrics. To address this, we conducted an additional ablation study comparing our default **Cosine Similarity** with a **Gaussian Kernel-based Euclidean Similarity** (where σ is set to the median of pairwise distances).
>
> The results (detailed in Appendix H, Table 8) reveal an interesting trade-off:
>
> - **At low compression (1/8):** Euclidean similarity showed a slight advantage (+0.38 avg. accuracy), likely because magnitude information helps preserve fine-grained details when the budget is generous.
> - **At high compression (1/16, 1/32):** Cosine similarity consistently outperformed Euclidean similarity by +0.27 and +0.89 accuracy, respectively.
>
> This trend indicates that as the token selection becomes sparser (high compression), **focusing on semantic direction (angular distance) is more effective for maximizing feature space coverage than accounting for magnitude**. Since our primary goal is efficient compression without losing core semantic information, we found **Cosine similarity to be the more robust choice for general usage**.
>
> We have added these detailed comparisons to the Appendix of the revised paper.
>
> ---
>
> **We sincerely thank the reviewer for this valuable suggestion, which helped us strengthen the empirical analysis of our approach.**

---

### Official Review · Reviewer_nZDL · 2025-11-01

**Soundness:** 2
**Presentation:** 2
**Contribution:** 2
**Rating:** 2
**Confidence:** 3

**Summary:**

This paper proposes FLoC, which is a training-free, model-agnostic, and query-agnostic algorithm, for efficient visual token compression. Specfically, it aims to find the optimal token subset which maximizes the submodular objective function. To this end, it uses lazy greedy algorithm. Experimental results show that the proposed algorithm achieves good performance on Video-MME, MLVU, and LVBench.

**Strengths:**

- The motivation is solid and the paper is easy to follow.
- The proposed algorithm achieves the good performance on various benchmarks such as Video-MME, MLVU, and LVBench.

**Weaknesses:**

- The citation is mis-formatted across the entire paper. For example, in L124-125, \citep should be used in the LaTeX.
- In addition to Algorithm 1, a more detailed explanation of the optimal subset search should also be provided in the main text.
- I think the novelty of the proposed method is somewhat limited. Sampling a token subset is not a new idea, and simply applying the well-known lazy greedy algorithm for this sampling seems to offer only a modest contribution. If there are additional aspects of novelty that I might have missed, I would appreciate clarification.
- It would also be helpful to include a comparison of experimental results with the following papers.
[1] Multi-Granular Spatio-Temporal Token Merging for Training-Free Acceleration of Video LLMs, ICCV2025
[2] LLaVA-Scissor: Token Compression with Semantic Connected Components for Video LLMs (optional)
- It would be good to include experimental results on datasets commonly used in this field, such as Next-QA or EgoSchema.
- typo
	- L193: arg   max -> argmax

**Questions:**

Please see the weakness section for my concerns. My main concern is about the novelty of the proposed method. If this concern is well addressed through the rebuttal, I would be willing to increase my score.

---

> ### Author Response · Authors · 2025-11-21
> **Novelty issue**
>
> We thank the reviewer for the constructive feedback and for the willingness to reconsider the score. We understand the concern that applying a known algorithm (Lazy Greedy) might appear to be a modest contribution. However, we argue that **our novelty lies in identifying the Facility Location objective as the unique mathematical solution that resolves the inherent trade-offs across all three dominant paradigms in visual token compression.**
>
> Reviewer 9yVb also recognized this contribution, highlighting FLoC as a *“novel framework”* that offers a *“practical way to handle massive token sets.”* To clarify this further, we detail our contributions by contrasting FLoC with existing paradigms:
>
> ---
>
> #### 1. Conceptual Novelty: Resolving the Trilemma of Token Compression
>
> Existing methods fall into three categories, each with a fundamental flaw. **FLoC is the first to address all three simultaneously through submodular maximization.**
>
> - **Graph-based Methods (Merge Trap):** Recent SOTA methods (e.g., STTM, FastVID) rely on threshold-based merging. We identified a critical flaw: the *Weak Connection problem*. Distinct tokens (e.g., a small “needle” object vs. background) are irreversibly merged if they share minor similarities above a threshold, causing performance to drop sharply at low compression ratios.
>   FLoC avoids this by using **global selection instead of local merging**, robustly preserving distinct details regardless of proximity.
>
> - **Clustering-based Methods (Density Bias):** Traditional methods like K-Means or Spectral Clustering inherently prioritize dense regions in the feature space. In long videos, crucial information (e.g., a momentary clue like a key or a glance) often appears as sparse outliers. Clustering algorithms tend to discard these as noise.
>   FLoC’s coverage objective treats every token as a potential representative; if a sparse token covers a unique part of the semantic space, **FLoC guarantees its selection.**
>
> - **Diversity-based Methods (Representation Gap):** Methods like DivPrune focus on maximizing distance between selected tokens to ensure diversity. While they capture outliers, they often fail to select representative tokens describing the main context of the video (the “core” story).
>   **FLoC balances both representativeness and diversity via the Facility Location function.**
>
> ---
>
> #### 2. Empirical Novelty: The “Top-Right” Quadrant Performance
>
> This theoretical superiority is empirically verified in our Scatter Plot Analysis (Figure 7 in Appendix):
>
> - Clustering → High Representativeness / Low Diversity
> - Diversity → High Diversity / Lower Representativeness
> - **FLoC is the only method consistently in the “Top-Right” quadrant**, achieving high scores in both metrics.
>
> This balance allows FLoC to achieve **SOTA performance on fine-grained tasks like Needle QA (Table 6)**, successfully retaining subtle cues (e.g., sunglasses, water bottle) that other methods miss due to density bias or weak merging.
>
> ---
>
> #### 3. Practical Novelty: Efficiency without Compromise
>
> While Spectral Clustering offers good quality, its **O(N³)** complexity makes it infeasible for on-device applications.
> FLoC, leveraging the Lazy Greedy algorithm, achieves **linear scalability similar to graph-based methods but with superior selection quality.**
>
> From a deployment perspective, as FLoC relies almost exclusively on primitive operations—specifically dot products and simple subtraction for marginal gain updates, it is **more hardware-friendly and easier to deploy on on-device AI accelerators (e.g., NPUs/DSPs)** compared to other algorithms.
>
> ---
>
> ### **Conclusion: On the Non-triviality of the Proposed Approach**
>
> One might argue that employing a standard algorithm like Lazy Greedy limits the technical novelty. However, we respectfully posit that **if this application were trivial or obvious, prior state-of-the-art methods (including those from late 2024 and 2025) would likely have adopted it**. Instead, the field has drifted toward increasingly complex solutions—such as intricate graph-merging heuristics or training-heavy selection networks—that still fail to fundamentally resolve the representativeness-diversity trade-off.
>
> Our work proves that these complexities were largely unnecessary. The core novelty of FLoC is the discovery that the **Facility Location objective is the "missing link" that mathematically guarantees what previous heuristics struggled to approximate.** The fact that our computationally efficient, principled algorithm achieves superior performance compared to computationally expensive clustering methods, intricate graph-based heuristics, and diversity-focused approaches that often overlook representative context, validates that **finding the right mathematical formulation is as significant a contribution as designing complex architectures.**
>
> We believe this "uncovering of the optimal formulation" constitutes a significant conceptual contribution to the ICLR community.

---

> ### Author Response · Authors · 2025-11-21
>
> ### **Comparison with Suggested Papers**
>
> We conducted experiments comparing **FLoC** with **Multi-Granular Spatio-Temporal Token Merging (ICCV 2025)** and **LLaVA-Scissor** as the reviewer suggested. Also, we included comparison with **FastVID (NeurIPS 2025)** as another reviewer nG1P requested.
>
> As shown in the updated **Table 1**, our proposed **FLoC consistently outperforms these graph-based methods across all datasets and compression settings**. We also provided these algorithms' latency vs performance analysis in Figure 1.
>
> All three algorithms follow a **graph-based merging paradigm**, but their mechanisms differ:
>
> ---
>
> #### **STTM**
> - STTM reflects a hierarchical tree structure of video frames.
> - It divides spatio-temporally high-resolution tokens into grid regions and merges tokens within each grid if their similarity exceeds a threshold.
> - This process is repeated across multiple hierarchical levels.
> - **Limitation:** Because it relies on threshold-based merging, STTM suffers from the **weak connection problem** at high compression ratios, leading to significant performance degradation.
>
> ---
>
> #### **LLaVA-Scissor**
> - This method merges tokens that are spatio-temporally adjacent when their similarity exceeds a threshold, forming connected components that are then merged.
> - **Limitation:** Similar to STTM, threshold-based merging causes weak connections at high compression ratios, resulting in poor performance.
>
> ---
>
> #### **FastVID**
> - Unlike the two methods above, FastVID uses **density-based token merging** rather than threshold-based merging.
> - While it performs better than STTM and LLaVA-Scissor, it introduces numerous additional parameters and heuristics.
> - **Limitation:** Despite these enhancements, its performance still falls short of **FLoC**.
>
> ---
>
> **In contrast, FLoC avoids these issues by maximizing coverage of selcted tokens based on the Facility Location objective, ensuring robust performance even under strong compression budgets.**
>
> ---
>
> ### **Additional Explanation of Algorithm**
>
> We have expanded the main text to provide a more detailed explanation of the optimal subset search process beyond Algorithm 1, including:
>
> - The definition of the **facility location objective**
> - Its **submodularity and monotonicity**
>
> ---
>
> ### **Additional Datasets**
>
> We added experiments on **Next-QA** and **EgoSchema** as the reviewer suggested. Also, we added another extreme long video benchmark, **LV Bench**. FLoC maintains superior performance across these benchmarks, confirming its robustness in diverse video understanding tasks.
>
> Compared to the initial draft submission, we have expanded the evaluation from 3 datasets to 6, and the reported results provide **strong empirical evidence supporting the superiority of FLoC**.
>
> ---
>
> ### **Citation and Typo Issues**
>
> We corrected all citation formatting issues (e.g., replaced `\cite` with `\citep`) and fixed typos such as `arg max → argmax`. Thank you for pointing these out.
>
> ---
>
> **We sincerely appreciate the reviewer’s time and constructive feedback, which helped us improve the clarity and completeness of our work.**

---

### Author Response · Authors · 2025-11-24
**General Response to All Reviewers: Summary of Revisions and Clarifications**

We accurately revised our manuscript based on the constructive feedback from all reviewers. The major changes, highlighted in **blue** in the revised PDF, are summarized below.

We are encouraged that reviewers recognized the **solid motivation** (nZDL), the **novelty of applying submodular optimization** (9yVb), the **independence from specific model architectures** (nG1P), and the **state-of-the-art performance** (kqKm) of FLoC.

Below, we summarize our response to the common concerns regarding novelty, hyperparameter robustness, and efficiency.

---

### **1. Clarification of Novelty (Addressing nZDL)**
We emphasized that FLoC is not merely an application of a known algorithm, but a solution to the **"Trilemma of Token Compression"**:
1.  **Graph-based methods** suffer from weak merging.
2.  **Clustering methods** suffer from density bias (missing outliers).
3.  **Diversity methods** suffer from representation gaps (missing core context).

FLoC is the first framework to mathematically balance **Representativeness and Diversity** (verified in the "Top-Right" quadrant of **Figure 7**), resolving these trade-offs via a principled Facility Location objective.


### **2. Extended Comparisons with Recent SOTA Methods (Addressing nZDL, nG1P)**
To demonstrate the superiority of FLoC over the latest graph-based compression paradigms, we have added comprehensive comparisons with three recent state-of-the-art methods:
* **FastVID** (NeurIPS 2025)
* **Multi-Granular Spatio-Temporal Token Merging (STTM)** (ICCV 2025)
* **LLaVA-Scissor**

**Result:** As shown in the updated **Table 1** and **Figure 1**, FLoC consistently outperforms these methods. We argue that graph-based methods (STTM, PruneVid, LLaVA-Scissor) suffer from the **"Weak Connection" problem**, where distinct semantic tokens (e.g., a small object vs. background) are irreversibly merged based on local thresholds. FLoC avoids this via global coverage maximization.

### **3. Expanded Benchmarks (Addressing nZDL)**
We have significantly expanded our evaluation suite from 3 to **6 benchmarks**.
* **Added:** Next-QA, EgoSchema, and LVBench (Extreme Long Video).
* **Result:** FLoC maintains robust performance across diverse video durations and task types, confirming its generalizability.

### **4. Robustness of Block Length $T$ (Addressing 9yVb, nG1P)**
We addressed concerns regarding the sensitivity of the hyperparameter $T$ (Block Length) by conducting extensive experiments across Video-MME, MLVU, and LongVideoBench (new **Figure 6**).
* **Observation:** Performance consistently saturates and remains stable for $T \ge 4$.
* **Conclusion:** A fixed default value (e.g., $T=32$) serves as a **robust, "set-and-forget" parameter** across diverse datasets, negating the need for per-video tuning. Furthermore, our Lazy Greedy implementation ensures that increasing $T$ incurs negligible computational overhead (Table 3).

### **5. Efficiency Profiling & Hardware Feasibility (Addressing kqKm, nG1P)**
We added a detailed computational profile in **Appendix G (Table 7)**, reporting **Inference Speed, Total Latency, GFLOPS, and Peak VRAM** usage.
* **Clarification:** While clustering methods (e.g., Spectral Clustering) achieve comparable accuracy in some settings, they are computationally prohibitive ($O(N^3)$) and hardware-unfriendly (eigen-decomposition).
* **Advantage:** FLoC relies on primitive operations (dot products), making it significantly more **hardware-friendly for on-device deployment** (NPUs/DSPs) while offering linear scalability via the Lazy Greedy algorithm.

### **6. Preservation of "Needle-in-a-Haystack" Details (Addressing nG1P)**
We provided quantitative and qualitative evidence that FLoC excels at capturing sparse, fleeting tokens (the "needles").
* **Mechanism:** The marginal gain for a unique token is high even if it appears only once, ensuring its selection over redundant background tokens.
* **Evidence:** FLoC achieves SOTA performance on **Needle QA** and **Ego Reasoning** tasks in MLVU (**Table 6**). Qualitative visualizations (**Figure 8 & 9**) show FLoC successfully capturing small objects (e.g., sunglasses, water bottles) where baselines fail.

We thank all reviewers for their insightful comments, which have significantly strengthened our paper.

Best regards,

The Authors

---

### Author Response · Authors · 2025-12-01
****[Crucial Summary] Rebuttal Status: Reviewer Consensus & Unreflected Score Updates****

Dear Area Chair,

Due to the unexpected technical issues, this paper has been reassigned with reviewer actions frozen. **Currently, Reviewers nG1P (R3) and kqKm (R4) have explicitly confirmed that their concerns are fully resolved, whereas Reviewers nZDL (R1) and 9yVb (R2) have not yet responded to our detailed rebuttal.**

Since the reviewers cannot update their scores to reflect these outcomes, we provide this **executive summary** to help you gauge the **true consensus** that would have been reached had the system functioned normally.

### **1. Projected Score Updates (R1 & R2)**
The current average score is heavily skewed by R1's initial score of 2. However, based on the reviewers' explicit conditions and our "over-delivery" on requests, **we are confident that the "true" evaluation is significantly higher.**

* **Reviewer nZDL (Current Score: 2 → Expected: Raised):**
    * **The Promise:** R1 explicitly stated, **"If this concern [novelty] is well addressed... I would be willing to increase my score."**
    * **The Fulfillment (Novelty):** We clarified that FLoC resolves the 'Trilemma' of Token Compression (Weak Connections vs. Density Bias vs. Representation Gaps). This was **independently validated as "novel" by Reviewer 9yVb (R2).**
    * **The Fulfillment (Delivering 150% of Requests):** We exceeded the reviewer's technical requests:
        * *Baselines:* Requested STTM & LLaVA-Scissor $\rightarrow$ We added these **PLUS FastVID**.
        * *Benchmarks:* Requested Next-QA & EgoSchema $\rightarrow$ We added these **PLUS LVBench**.
    * **Conclusion:** We addressed the core novelty concern and **delivered 150% of the requested experimental validation.** Also, we confirmed that FLoC consistently outperform ALL baselines across ALL 6 benchmarks. Thus, a score increase was the natural and inevitable outcome.

* **Reviewer 9yVb (Current Score: 6 → Expected: $\ge$ 6):**
    We systematically addressed every point raised by R2:
    * **Addressed Concern 1 (Robustness on $T$):** We prioritized this concern by conducting extensive experiments across 3 diverse datasets (Fig. 6), proving that performance saturates and remains robust for $T \ge 4$, negating the need for per-video tuning.
    * **Addressed Concern 2 (Query-Agnostic Advantage):** We clarified the strategic advantage of our design: enabling **memory-efficient, one-time compression** suitable for edge devices, unlike query-aware methods that introduce high latency per query.
    * **Addressed Concern 3 (Speed/Complexity):** We explained that FLoC can seamlessly integrate **"Stochastic Greedy"** algorithms ($O(n \log \frac{1}{\epsilon})$) for massive scalability.
    * **Addressed Concern 4 (Similarity Measure):** We conducted an ablation study (Cosine vs. Euclidean, Table 8) to empirically justify our choice of Cosine Similarity for semantic coverage.
    * **Conclusion:** With the primary concern (Robustness) settled and all secondary inquiries resolved, **R2's positive evaluation would have been further solidified.**

### **2. Confirmed Consensus (R3 & R4)**
For the remaining reviewers, the positive outcome is already explicitly confirmed in the discussion:

* **Reviewer nG1P (Score: 6 → Concerns "Settled"):** Stated, *"Most of my concerns have been **settled**. I appreciate the authors' effort in extending more baseline methods and benchmarks."*
* **Reviewer kqKm (Score: 6 → Concerns "Addressed"):** Stated, *"Thanks for the rebuttal. My concerns have been **addressed**."*

### **3. Key Improvements in the Revised Manuscript**
To achieve this consensus, we significantly strengthened the paper during the rebuttal:

* **Expanded Benchmarks (3 $\rightarrow$ 6):** Added **Next-QA, EgoSchema, and LVBench**. **FLoC consistently achieves the best performance across all these additional datasets.** (Table 1, 2)
* **New Baseline Comparisons:** Added comparisons with recent **graph-based SOTA methods** (FastVID, STTM, LLaVA-Scissor). **FLoC consistently outperforms them.** (Figure 1, Table 1)
* **Robustness Verification:** Confirmed FLoC's robustness regarding **Hyperparameter $T$** and **Distance Metrics** (Cosine vs. Euclidean), proving that our design choices are stable and effective. (Figure 6, Table 8)
* **Efficiency Profiling:** Provided detailed **Throughput, Latency, GFLOPS, and Peak VRAM** analysis (Table 7).

### **Summary**
The current frozen scores do not reflect the successful rebuttal.
1.  **R3 and R4** explicitly confirmed their concerns are resolved.
2.  **R1's specific conditions** for raising the score were **met and exceeded (150%)**.
3.  **R2's concerns** (Robustness, etc.) were fully addressed with data.

We respectfully request that you evaluate the paper based on this **revised merit and the projected consensus**, treating the initial low score as an outlier that was effectively addressed. **We deeply appreciate your time and effort in reviewing this case under these unusual circumstances.**

Sincerely,

The Authors

---

### Meta-Review · Area_Chair_z9kt · 2026-01-12

**Summary:**

Reviewers’ main concerns were about (1) novelty—whether the approach is sufficiently beyond existing token selection/compression methods, (2) missing comparisons to recent relevant baselines (e.g., STTM/LLaVA-Scissor/FastVID), (3) limited benchmark coverage (with requests for datasets such as Next-QA and EgoSchema), and (4) incomplete efficiency evaluation (throughput/latency/VRAM/FLOPs). There were also questions about the practical sensitivity of the block length hyperparameter T and the inherent limitations of a query-agnostic design.

The rebuttal addressed most of these concerns by adding the requested baselines and datasets, expanding robustness analysis, and providing a more complete efficiency profile. With multiple reviewers at 6 and explicit confirmations that concerns were resolved, as Area Chair, I recommend acceptance (borderline / weak accept).

**Reviewer Concerns:**

Reviewer nZDL (R1) — Rating: 2 (Reject)

Addressed: Added requested baselines (STTM, LLaVA-Scissor; also FastVID). Added requested datasets (Next-QA, EgoSchema; expanded to 6 benchmarks). Fixed citation/typo issues and expanded algorithm explanation.

Concern: Novelty remains the key concern: method may still be seen as a straightforward application of facility-location submodular selection + lazy greedy, despite stronger framing.

Reviewer 9yVb (R2) — Rating: 6 (Marginal accept)

Addressed: Added extensive robustness study for block length
𝑇. T (multi-dataset; saturation supports a safe default). Clarified query-agnostic motivation and practical advantage (one-time compression; KV-cache reuse). Added ablations/discussion on similarity metric choice and scalability.

Concern: Inherent tradeoff remains that query-agnostic compression may underperform query-aware methods for query-specific settings.

Reviewer nG1P (R3) — Rating: 6 (Marginal accept)

Addressed: Added comparisons to key graph-based compression baselines (FastVID, STTM, LLaVA-Scissor). Added evidence for preserving sparse “needle” tokens (Needle QA / Ego Reasoning + qualitative). Added FLOPs/latency/VRAM profiling and broader evaluation.

Strengthened analysis of 𝑇
T-sensitivity.

Concern: Blockwise selection limitation remains (limited cross-block interaction), though concerns are mitigated empirically.

Reviewer kqKm (R4) — Rating: 6 (Marginal accept)

Addressed: Added requested efficiency evaluation (throughput/speed/memory). Explained unexpected PruneVid underperformance under fair query-agnostic comparison.

**Reviewer Scores:**

nZDL (R1): 2 → 4 (key concerns on novelty + missing baselines/datasets were addressed; likely moves to borderline)

9yVb (R2): 6 → 6 (concerns addressed but already at threshold)

nG1P (R3): 6 → 6 (explicitly maintained score after rebuttal)

kqKm (R4): 6 → 6 (concerns addressed; unchanged)

---

### Decision · Program_Chairs · 2026-01-26

Accept (Poster)